# Complex behavior from intrinsic motivation to occupy future action-state path space

Jorge Ramírez-Ruiz [1] ✉, Dmytro Grytskyy[1], Chiara Mastrogiuseppe [1], Yamen Habib [1] & Rubén Moreno-Bote[1,2]

Most theories of behavior posit that agents tend to maximize some form of reward or utility. However, animals very often move with curiosity and seem to be motivated in a reward-free manner. Here we abandon the idea of reward maximization and propose that the goal of behavior is maximizing occupancy of future paths of actions and states. According to this maximum occupancy principle, rewards are the means to occupy path space, not the goal per se; goal-directedness simply emerges as rational ways of searching for resources so that movement, understood amply, never ends. We find that action-state path entropy is the only measure consistent with additivity and other intuitive properties of expected future action-state path occupancy. We provide analytical expressions that relate the optimal policy and state-value function and prove convergence of our value iteration algorithm. Using discrete and continuous state tasks, including a high-dimensional controller, we show that complex behaviors such as "dancing", hide-and-seek, and a basic form of altruistic behavior naturally result from the intrinsic motivation to occupy path space. All in all, we present a theory of behavior that generates both variability and goal-directedness in the absence of reward maximization.

Natural agents are endowed with a tendency to move, explore, and interact with their environment[1,2]. For instance, human newborns unintentionally move their body parts[3], and 7–12-month-old infants spontaneously babble vocally[4] and with their hands[5]. Exploration and curiosity are major drives for learning and discovery through information-seeking[6–8]. These behaviors seem to elude a simple explanation in terms of extrinsic reward maximization. However, intrinsic motivations, such as curiosity, push agents to visit new states by performing novel courses of action, which helps learning and the discovery of even larger rewards in the long run[9,10]. Therefore, it has been argued that exploration and curiosity could arise as a consequence of seeking extrinsic reward maximization by endowing agents with the necessary inductive biases to learn in complex and ever-changing natural environments[11,12].

While most theories of rational behavior do posit that agents are reward maximizers[13–16], very few of us would agree that the sole goal of living agents is maximizing money gains or food intake.

Indeed, expressing excessive emphasis on these types of goals is usually seen as a sign of psychological disorders[17,18]. Further, setting a reward function by design as the goal of artificial agents is, more often than not, arbitrary[14,19–21], leading to the recurrent problem faced by theories of reward maximization of defining what rewards are[22–26]. In some cases, like in artificial games, rewards can be unambiguously defined, such as number of collected points or wins[27]. However, in most situations defining rewards is task-dependent, non-trivial, and problematic. For instance, a vacuum cleaner robot could be designed to either maximize the weight or volume of dust collected, energy efficiency, or a weighted combination of them[28]. In more complex cases, companies can aim at maximizing profit, but without a suitable innovation policy, profit maximization can be self-defeating[29]. Even when a task is well-defined from the perspective of an external observer, the actions and states of natural agents are not always compatible with a deterministic maximization of rewards and are consistently found to be highly variable[30–32]. Natural agents are

[1]Center for Brain and Cognition, Departament d'Enginyeria i Escola d'Enginyeria, Universitat Pompeu Fabra, Barcelona, Spain. [2]Serra Húnter Fellow Programme, Universitat Pompeu Fabra, Barcelona, Spain. ✉e-mail: jorgeerrz@gmail.com

also sensitive to the uncertainty in or about the environment, behaving in ways that are inconsistent with strict reward maximization in familiar environments[33,34]. While there are isolated attempts to understand the function and mechanisms of behavioral variability[35], and risk sensitivity in uncertain environments[36–38], general principles for the importance of stochasticity in so-called goal-directed behavior are lacking.

Here, we abandon the idea that the goal is maximizing extrinsic rewards and that movement over space is a means to achieve this goal. Instead, we adopt the opposite view, inspired by the nature of our intrinsic drives: we propose that the objective is to maximally occupy action-state path space, understood in a broad sense, in the long term. We call this principle the maximum occupancy principle (MOP), which posits that the goal of agents is to generate all sorts of variable behaviors and visit, on average, as much space (action-state paths) as possible in the future. According to MOP, extrinsic rewards serve to obtain the energy necessary to move in order to occupy action-state space, they are not the goals per se. The usual exploration-exploitation tradeoff[39], therefore, disappears: agents that seek to occupy space "solve" this issue naturally because they care about rewards only as a means to an end. Furthermore, in this sense, surviving is only preferred because it is needed to keep visiting action-state path space. Our theory provides a rational account of exploratory and curiosity-driven behavior where the problem of defining a reward function vanishes, and captures the variability of behavior[40–45] by taking it as a principle.

In this work, we model a MOP agent interacting with the environment as a Markov decision process (MDP) where the intrinsic, immediate reward is the occupancy of the next action-state visited, which is largest when performing an uncommon action and visiting a rare state—there are no extrinsic rewards (i.e., no task is defined) that drive the agent. We show that (weighted) action-state path entropy is the only measure of occupancy consistent with additivity per time step, positivity, and smoothness. Due to the additivity property, the value of being in a state, defined as the expected future time-discounted action-state path entropy, can be written in the form of a Bellman equation, which has a unique solution that can be found with an iterative map. Following this entropy objective leads to agents that seek variability while being sensitive to the constraints imposed by the agent-environment interaction on the future path availability. We demonstrate in various simulated experiments with discrete and continuous state and action spaces that MOP generates complex behaviors that, to the human eye, look genuinely goal-directed and playful, such as hide-and-seek in a prey-predator problem, dancing of a cartpole, a basic form of altruism in an agent-and-pet example, and rich behaviors in a high-dimensional quadruped.

MOP builds over an extensive literature on entropy-regularized reinforcement learning (RL)[46–56] or pure entropic objectives[57–62]. This body of work emphasizes the regularization benefits of entropy for learning, but extrinsic rewards still serve as the major drive of behavior, and arbitrary mixtures of action-state entropy are rarely considered[56]. Our work also relates to reward-free theories of behavior. These typically minimize prediction errors[63–68], seek novelty[69–71], or maximize data compression[72], and therefore the major behavioral driver depends on the agent's experience with the world. On the other hand, MOP agents find the action-states that lead to high future occupancy "interesting", regardless of experience. There are two other approaches that sit closer to this description, one maximizing mutual information between actions and future states (empowerment, MPOW)[20,73–75], and the other minimizing the distance between the actual and a desired state distribution (free energy principle, FEP)[76,77]. We show that both MPOW and FEP tend to collapse to deterministic policies with little behavioral variability. In contrast, MOP results in lively and seemingly goal-directed behavior by taking behavioral variability and the constraints of embodied agents as principles.

## Results

### Maximum occupancy principle

We model an agent as a finite action-state MDP in discrete time. The policy $\pi$ describes the probability $\pi(a|s)$ of performing action $a$, from some set $\mathcal{A}(s)$, given that the agent is in state $s$ at some time step, and $p(s'|s,a)$ is the transition probability from $s$ to a successor state $s'$ in the next time step given that action $a$ is performed. Starting at $t = 0$ in state $s_0$, an agent performing a sequence of actions and experiencing state transitions $\tau \equiv (s_0, a_0, s_1,...,a_t, s_{t+1},...)$ gets a return defined as

$$R(\tau) = \sum_{t=0}^{\infty} \gamma^t R(s_t, a_t) = -\sum_{t=0}^{\infty} \gamma^t \ln\left(\pi^\alpha(a_t|s_t) p^\beta(s_{t+1}|s_t, a_t)\right), \quad (1)$$

with action and state weights $\alpha > 0$ and $\beta \geq 0$, respectively, and discount factor $0 < \gamma < 1$. A larger return is obtained when, from $s_t$, a low-probability action $a_t$ is performed and followed by a low-probability transition to a state $s_{t+1}$. Therefore, maximizing the return in Eq. (1) favors "visiting" action-states $(a_t, s_{t+1})$ with a low transition probability. From $s_{t+1}$, another low-probability action-state transition is preferred, and so on, such that low-probability trajectories $\tau$ are more rewarding than high-probability ones. Thus, the agent is pushed to visit action-states that are rare or "unoccupied", implementing the intuitive notion of MOP. Due to the freedom to choose action $a_t$ given state $s_t$ and the uncertainty of the resulting next state $s_{t+1}$, apparent in Eq. (1), the term "action-states" used here is more natural than "state-actions". We stress that this return is purely intrinsic, namely, there is no extrinsic reward that the agent seeks to maximize. We define intrinsic rewards as any reward signal that depends on the policy or the state transition probability, and therefore it can change with the course of learning as the policy is improved, or the environment is learned. An extrinsic reward is the complementary set of reward signals: any function $R(s, a)$ that is both policy-independent and transition probability-independent, and therefore it does not change with the course of improving the policy or learning the state transition probability of the environment.

The agent is assumed to optimize the policy $\pi$ to maximize the state-value $V_\pi(s)$, defined as the expected return

$$V_\pi(s) \equiv \mathbb{E}_{a_t \sim \pi, s_{t+1} \sim p}[R(\tau)|s_0 = s]$$
$$= \mathbb{E}_{a_t \sim \pi, s_{t+1} \sim p}\left[\sum_{t=0}^{\infty} \gamma^t (\alpha\mathcal{H}(A|s_t) + \beta\mathcal{H}(S'|s_t, a_t))|s_0 = s\right] \quad (2)$$

given the initial condition $s_0 = s$ and following policy $\pi$, that is, the expectation is over the $a_t \sim \pi(\cdot|s_t)$ and $s_{t+1} \sim p(\cdot|s_t, a_t)$, $t \geq 0$. In the last identity, we have rewritten the expectations of the terms in Eq. (1) as a discounted and weighted sum of action and successor state conditional entropies $\mathcal{H}(A|s) = -\sum_a \pi(a|s) \ln \pi(a|s)$ and $\mathcal{H}(S'|s,a) = -\sum_{s'} p(s'|s,a) \ln p(s'|s,a)$, respectively, averaged over previous states and actions.

We define a MOP agent as the one that optimizes the policy to maximize the state value in Eq. (2). The entropy representation in Eq. (2) of MOP has several implications. First, agents prefer regions of state space that lead to a large number of successor states (Fig. 1a) or larger number of actions (Fig. 1b). Second, death (absorbing) states where only one action-state (i.e., "stay") is available forever are naturally avoided by a MOP agent, as they promise zero future action and state entropy (Fig. 1c—hence absorbing states $s^+$ have zero state-value, $V_\pi(s^+) = 0$). Therefore, our framework implicitly incorporates a survival instinct. Finally, regions of state space where there are "rewarding" states that increase the capacity of the agent to visit further action-states (such as filling an energy reservoir) are more frequently visited than others (Fig. 1d).

We found that maximizing the discounted action-state path entropy in Eq. (2) is the only reasonable way of formalizing MOP, as it is

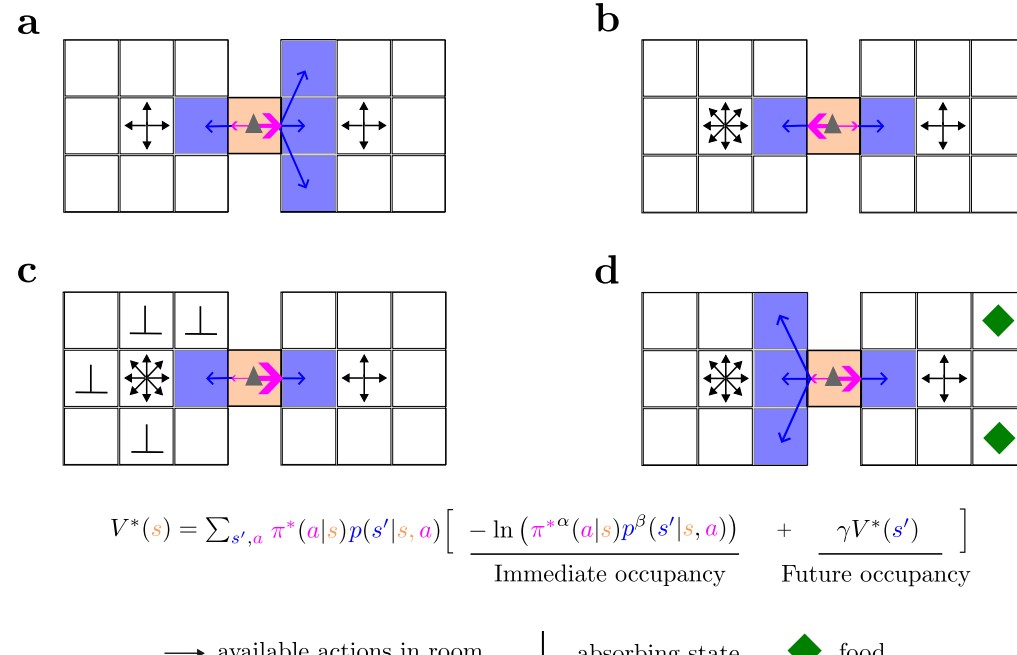

$$V^*(s) = \sum_{s',a} \pi^*(a|s)p(s'|s,a)\Big[ \underbrace{-\ln\big(\pi^{*\alpha}(a|s)p^\beta(s'|s,a)\big)}_{\text{Immediate occupancy}} + \underbrace{\gamma V^*(s')}_{\text{Future occupancy}} \Big]$$

⟶ available actions in room     ⊥ absorbing state     ◆ food

**Fig. 1 | MOP agents maximize action-state path occupancy. a** A MOP agent (gray triangle) in the middle of two rooms has the choice between going left or right. When the number of actions (black arrows) in each room is the same, the agent prefers going to the room with more state transitions (blue arrows indicate random transitions after choosing moving right or moving left actions, and pink arrow width indicates the probabilities of those actions). **b** When the state transitions are the same in the two rooms, the MOP agent prefers the room with more available actions. **c** If there are many absorbing states in the room where many actions are available, the MOP agent avoids it. **d** Even if there are action and state-transition incentives (in the left room), an MOP agent might prefer a region of state space where it can reliably get food (right room), ensuring occupancy of future action-state paths. See Supplemental Fig. D.1 for a more formal example.

the only measure of action-state path occupancy in Markov chains consistent with the following intuitive conditions (see Methods and Supplemental Sec. A): if a path $\tau$ has probability $p$, visiting it results in an occupancy gain $C(p)$ that (i) decreases with $p$ and (ii) is first-order differentiable. Condition (i) implies that visiting a low-probability path increases occupancy more than visiting a high-probability path, and our agents should tend to occupy "unoccupied" path space; condition (ii) requires that the measure should be smooth. We also ask that (iii) the occupancy of paths, defined as the expectation of occupancy gains over paths given a policy, is the sum of the expected occupancies of their subpaths (additivity condition). This last condition implies that agents can accumulate occupancy over time by keeping visiting low-probability action-states, but the accumulation should be consistent with the Markov property of the decision process. These conditions are similar but not exactly the same as those used to derive Shannon's information measure[78] (see Methods and Supplemental Sec. A).

### Optimal policy and state-value function

The state-value $V_\pi(s)$ in Eq. (2) can be recursively written using the values of successor states through the standard Bellman equation

$$V_\pi(s) = \alpha\mathcal{H}(A|s) + \beta\sum_a \pi(a|s)\mathcal{H}(S'|s,a) + \gamma\sum_{a,s'}\pi(a|s)p(s'|s,a)V_\pi(s')$$
$$= \sum_{a,s'}\pi(a|s)p(s'|s,a)\big(-\alpha\ln\pi(a|s) - \beta\ln p(s'|s,a) + \gamma V_\pi(s')\big),$$
$$(3)$$

where the sum is over the available actions $a$ from state $s$, $\mathcal{A}(s)$, and over the successor states $s'$ given the performed action at state $s$. The number of actions and state transitions available does not need to be the same for all states $s$. In particular, absorbing states $s^+$ have only the "stay" action available, leading to zero action and state transition entropies, and therefore they have zero state-value $V_\pi(s^+) = 0$ regardless of the policy.

The optimal policy $\pi^*$ that maximizes the state value is defined as $\pi^* = \arg\max_\pi V_\pi$ and the optimal state value is

$$V^*(s) = \max_\pi V_\pi(s), \qquad (4)$$

where the maximization is with respect to the $\{\pi(\cdot|\cdot)\}$ for all actions and states. To obtain the optimal policy, we first determine the critical points of the expected return $V_\pi(s)$ in Eq. (3) using Lagrange multipliers (see Methods and Supplemental Sec. B). The optimal state-value $V^*(s)$ is found to obey the non-linear self-consistency set of equations

$$V^*(s) = \alpha\ln Z(s) = \alpha\ln\left[\sum_a \exp\left(\alpha^{-1}\beta\mathcal{H}(S'|s,a) + \alpha^{-1}\gamma\sum_{s'}p(s'|s,a)V^*(s')\right)\right],$$
$$(5)$$

where $Z(s)$ is the partition function, defined by substitution, and the critical policy satisfies

$$\pi^*(a|s) = \frac{1}{Z(s)}\exp\left(\alpha^{-1}\beta\mathcal{H}(S'|s,a) + \alpha^{-1}\gamma\sum_{s'}p(s'|s,a)V^*(s')\right). \qquad (6)$$

Note that the optimal policy in MOP only depends on a single parameter, the ratio $\beta/\alpha$ and that the optimal state-value function is simply scaled by $\alpha$. We find that the solution to the non-linear system of Eqs. (5) is unique and, moreover, the unique solution is the absolute maximum of the state-values over all policies (Supplemental Sec. C).

To determine the actual value function from such non-linear set of equations, we derive an iterative map, a form of value iteration that exactly incorporates the optimal policy at every step. Defining $z_i = \exp(\alpha^{-1}\gamma V(s_i))$, $p_{ijk} = p(s_j|s_i, a_k)$ and $\mathcal{H}_{ik} = \alpha^{-1}\beta\mathcal{H}(S'|s_i,a_k)$, Eq. (5)

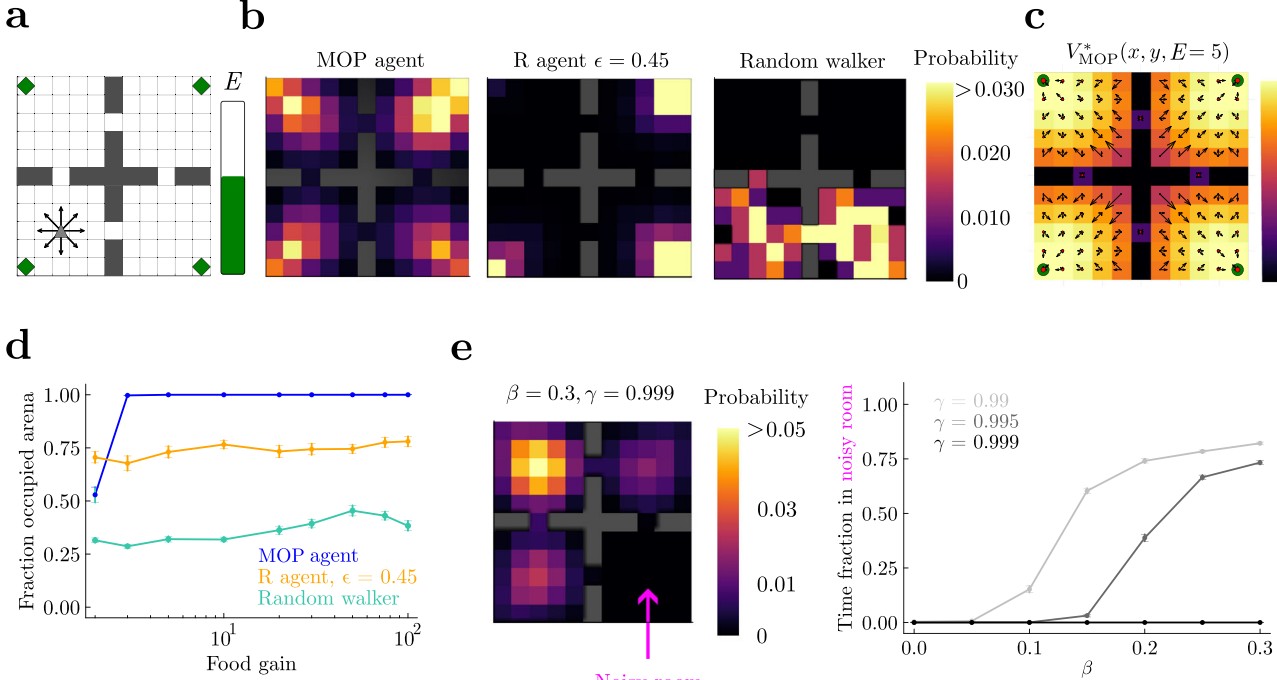

**Fig. 2 | Maximizing future path occupancy leads to high occupancy of physical space. a** Grid-world arena. The agents have nine available actions (arrows, and staying still) when alive (internal energy $E$ larger than zero) and away from walls. There are four rooms, each with a small food source in a corner (green diamonds). **b** Probability of visited spatial states for an MOP agent, an $\epsilon$-greedy reward (R) agent that survives as long as the MOP agent, and a random walker. Food gain = 10 units, maximum reservoir energy = 100, episodes of $5 \times 10^4$ time steps, and $(\alpha, \beta) = (1, 0)$ for the MOP agent. All agents are initialized in the middle of the lower left room. **c** Optimal value function $V(s)$ over locations when energy is $E = 5$. Black arrows represent the optimal policy given by Eq. (6); their length is

proportional to the probability of each action. The size of red dots is proportional to the probability of the "stay" action. **d** Fraction of locations of the arena visited at least once per episode as a function of food gain. Error bars correspond to s.e.m over 50 episodes. **e** Noisy room problem. The bottom right room of the arena was noisy, such that agents in this room jumped randomly to neighboring locations regardless of their actions. Food gain equals maximum reservoir energy = 100. Histogram of visited locations for an episode as long as in (**b**) for a MOP agent with $\beta = 0.3$ (left) and time fraction spent in the noisy room (right) show that MOP agents with $\beta > 0$ can either be attracted to the room or repelled depending on $\gamma$.

can be turned into the iterative map

$$z_i^{(n+1)} = \left( \sum_k w_{ik} e^{\mathcal{H}_{ik}} \prod_j \left( z_j^{(n)} \right)^{p_{ijk}} \right)^\gamma \qquad (7)$$

for $n \geq 0$ and with initial conditions $z_i^{(0)} > 0$. Here, the matrix with coefficients $w_{ik} \in \{0, 1\}$ indicates whether action $a_k$ is available at state $s_i$ ($w_{ik} = 1$) or not ($w_{ik} = 0$), and $j$ extends over all states, with the understanding that if a state $s_j$ is not a possible successor from state $s_i$ after performing action $a_k$ then $p_{ijk} = 0$. We find that the infinite series $z_i^{(n)}$ defined in Eq. (7) converges to a finite limit $z_i^{(n)} \to z_i^\infty$ regardless of the initial condition in the positive first orthant, and that $V^*(s_i) = \alpha \gamma^{-1} \ln z_i^\infty$ is the optimal state-value function, which solves Eq. (5) (Supplemental Sec. C). Iterative maps similar to Eq. (7) have been studied before[46,79], subsequently shown to have uniqueness[80] and convergence guarantees[54,81] in the absence of state entropy terms. A summary of results and particular examples can be found in Supplemental Sec. D.

We note that in the definition of return in Eq. (2) we could replace the absolute action entropy terms $\mathcal{H}(A|s)$ by relative entropies of the form $-D_{\mathrm{KL}}(\pi(a|s)||\pi_0(a|s)) = \sum_a \pi(a|s) \ln(\pi_0(a|s)/\pi(a|s))$, as in KL-regularization[46,50,55,79], but in the absence of any extrinsic rewards. In this case, one obtains an equation identical to (7) where the coefficients $w_{ik}$ are simply replaced by $\pi_0(a_k|s_i)$, one-to-one. This apparently minor variation uncovers a major qualitative difference between absolute and relative action entropy objectives: as $\sum_k w_{ik} \geq 1$, absolute entropy-seeking favors visiting states with a large action accessibility, that is, where the sum $\sum_k w_{ik}$ and thus the argument of Eq. (7) tends to

be largest. In contrast, as $\sum_k \pi_0(a_k|s_i) = 1$, maximizing relative entropies provides no preference for states $s$ with a large number of accessible actions $|\mathcal{A}(s)|$. This happens even if the default policy is uniform in the actions, as then the immediate intrinsic return becomes $-D_{\mathrm{KL}}(\pi(a|s)||\pi_0(a|s)) = \mathcal{H}(A|s) - \ln|\mathcal{A}(s)|$, instead of $\mathcal{H}(A|s)$. The negative logarithm penalizes visiting states with large number of actions, which is the opposite goal to occupying action-state path space (see details in Supplemental Sec. F).

## MOP agents quickly fill physical space

In very simple environments with high symmetry and little constraints, like open space, maximizing path occupancy amounts to performing a random walk that chooses at every step any available action with equal probability. However, in realistic environments where space is not homogeneous, where there are energetic limitations for moving, or where there are absorbing states, a random walk is no longer optimal. To illustrate how interesting behaviors arise from MOP in these cases, we first tested how a MOP agent moving in a 4-room and 4-food-sources environment (Fig. 2a) compares in occupying physical space to a random walker (RW) and to a reward-seeking agent (R agent) (see Methods for definitions of the agents). The definitions of the three agents are identical in most ways. They have nine possible movement actions, including not moving; they all have an internal state corresponding to the available energy, which reduces by one unit at every time step and gets increased by a fixed amount (food gain) whenever a food source is visited; and they can move as long as their energy is non-zero. The total state space is the Cartesian product between physical space and internal energy. The agents differ however in their objective

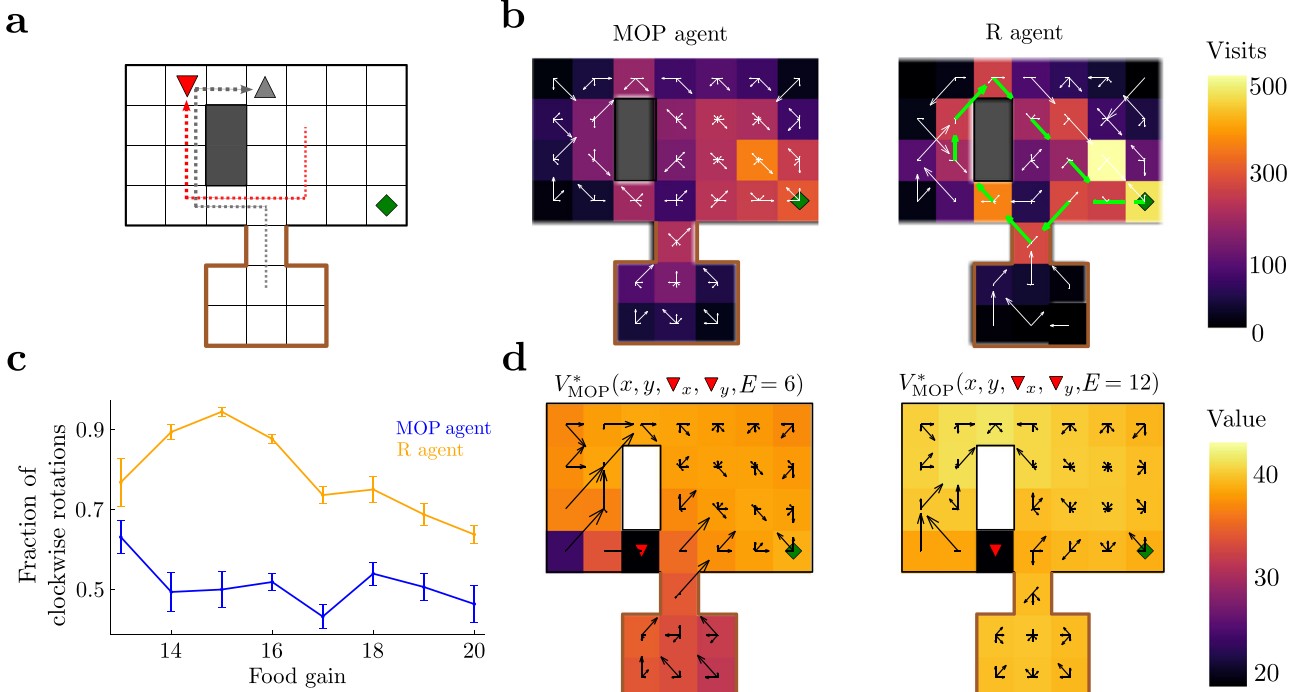

**Fig. 3 | Complex hide-and-seek and escaping strategies in a prey-predator example. a** Grid-world arena. The agent has nine available actions when alive and far from walls. There is a small food source in a corner (green diamond). A predator (red, down triangle) is attracted to the agent (gray, up triangle), such that when they are at the same location, the agent dies. The predator cannot enter the locations surrounded by the brown border. Arrows show a clockwise trajectory. **b** Histogram of visited spatial states across episodes for the MOP and R agents. The vector field at each location indicates probability of transition at each location. Green arrows on R agent show major motion directions associated with its dominant clockwise rotation. **c** Fraction of clockwise rotations (as in (**a**)) to total rotations as a function of food gain, averaged over epochs of 500 timesteps. Error bars are s.e.m. **d** Optimal value functions for different energy levels and same predator position; black arrows indicate optimal policy, as in Fig. 2c.

function. The MOP agent has a reward-free objective and implements MOP by maximizing path action entropy, Eq. (2). In contrast, the R agent maximizes future discounted reward (in this case, food), and displays stochastic behavior through an $\epsilon$-greedy action selection, with $\epsilon$ matched to the survival of the MOP agent (Supplemental Sec. E and Fig. E.2a). Finally, the RW is simply an agent that in each state takes a uniformly random action from the available actions at that state.

We find that the MOP agent generates behaviors that can be dubbed goal-directed and curiosity-driven (Supplementary Movie 1). First, by storing enough energy in its reservoir, the agent reaches far, entering the four rooms in the long term (Fig. 2b, left panel), and visiting every location of the arena except when food gain is small (Fig. 2d, blue line). In contrast, the R agent lingers over one of the food sources for most of the time (Fig. 2b, middle panel; Supplementary Movie 1). Although its $\epsilon$-greedy action selection allows for brief exploration of other rooms, the R agent does not on average visit the whole arena (Fig. 2d, orange line). Finally, the RW dies before it has time to visit a large fraction of the physical space (Fig. 2b, right panel). These differences hold for a large range of food gains (Fig. 2d). The MOP agent, while designed to generate variability, is also capable of deterministic behavior: when its energy is low, it moves toward the food sources with little to no variability, a distinct mark of goal-directedness (Fig. 2c, corner spots show that only one action is considered by optimal policy).

We next considered a slightly more complex environment where actions in one of the rooms lead to uniformly stochastic transitions to any of the neighboring locations (noisy room−a spatial version of the noisy TV problem[66,82]). A stochastic region in the environment can reflect uncertainty about this region (e.g., due to agents with limited resources) or "true" noise in the environment. Regardless of the source, a MOP agent with $\beta \neq 0$ will exhibit risk-sensitivity, i.e.,

preference or avoidance of this uncertainty[15]. In particular, MOP agents with $\beta > 0$ (see Eq. (2)) have a preference for stochastic state transitions and a priori they could get attracted and stuck in the noisy room, where actions do not have any predictable effect. Indeed, we see that for larger $\beta$, which measures the strength of the state entropy contribution to the agent's objective, the attraction to the noisy room increases (Fig. 2e, right panel). However, MOP agents also care about future states, and thus getting stuck in regions where energy cannot be predictably obtained is avoided by sufficiently long-sighted agents, as shown by the reduction of the time spent in the noisy room with increasing $\gamma$ (Fig. 2e; Supplemental Sec. E.3). This shows how MOP agents can tradeoff immediate with future action-state occupancy.

## Hide and seek in a prey-predator interaction

More interesting behaviors arise from MOP in increasingly complex environments. To show this, we next considered a prey and a predator in a grid world with a safe area (a "home") and a single food source (Fig. 3a). The prey (a "mouse", gray up triangle) is the agent whose behavior is optimized by maximizing future action path entropy, while the predator (a "cat", red down triangle) acts passively chasing the prey. The state of the agent consists of its location and energy level, but it also includes the predator's location being accurately perceived. The prey can move as in the previous 4-room grid world and it also has a finite energy reservoir. For simplicity, we only considered a food gain equal to the size of the energy reservoir, such that the agent fully replenishes its reservoir each time it visits the food source. The predator has the same available actions as the agent and is attracted to it stochastically, i.e., actions that move the predator towards the agent are more probable than those that move it away from it (Supplemental Sec. E.4).

MOP generates complex behaviors, not limited to visiting the food source to increase the energy buffer and hide at home. In

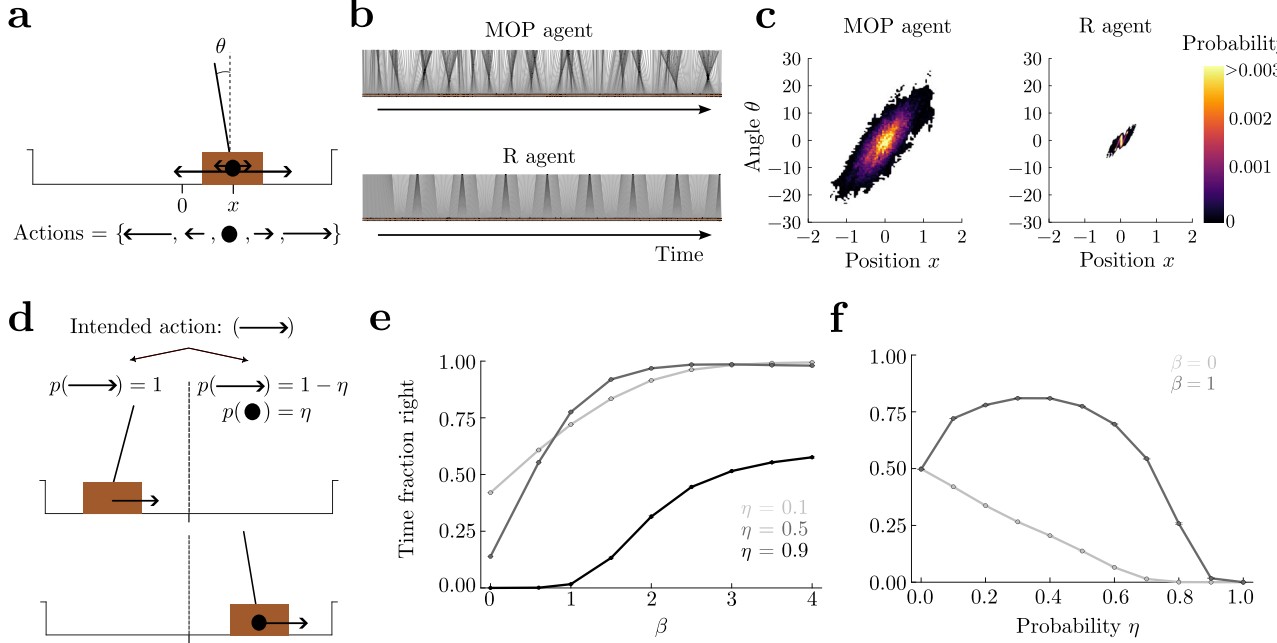

**Fig. 4 | Dancing of a MOP cartpole. a** The cart (brown rectangle) has a pole attached. The cartpole reaches an absorbing state if the magnitude of the angle $\theta$ exceeds 36° or its position reaches the borders. There are five available actions when alive: a big and a small force to either side (arrows on cartpole) and doing nothing (full circle). **b** Time-shifted snapshots of the pole in the reference frame of the cart as a function of time for the MOP (top) and R (bottom) agents. **c** Position and angle occupation for a $2 \times 10^5$ time step episode. **d** Here, the right half of the arena is stochastic, while the left remains deterministic. In the stochastic half, the intended state transition due to an applied action (force) succeeds with probability $1 - \eta$ (and thus zero force is applied with probability $\eta$). **e** Fraction of time spent on the right half of the arena increases as a function of $\beta$, regardless of the failure probability $\eta$. **f** The fraction has a non-monotonic behavior as a function of $\eta$ when state entropy is important for the agent ($\beta = 1$), highlighting a stochastic resonance behavior. When the agents do not seek state entropy ($\beta = 0$) the fraction of time spent by the agent on the right decreases with the failure probability, and thus they avoid the stochastic right side. $\gamma = 0.99$ for (**e**, **f**).

particular, the agent very often first teases the cat and then performs a clockwise rotation around the obstacle, which forces the cat to chase it around, leaving the food source free for harvest (Fig. 3a, arrows show an example; Supplementary Movie 2, MOP agent). Importantly, this behavior is not restricted to clockwise rotations, as the agent performs an almost equal number of counterclockwise rotations to free the food area (Fig. 3c, MOP agent, blue line). The variability of these rotations in the MOP agent is manifest in the lack of virtually any preferred directionality of movement in the arena at any single position. Indeed, arrows pointing toward several directions indicate that on average the mouse moves following different paths to get to the food source (Fig. 3b, MOP agent). Finally, the optimal value function and optimal policy show that the MOP agent can display deterministic behaviors as a function of internal state as well as distance to the cat (Fig. 3d): for instance, it prefers running away from the cat when energy is large (right), and it risks getting caught to avoid starvation if energy is small (left), both behaviors starkly opposite to stochastic actions.

The behavior of the MOP agent was compared with an R agent that receives a reward of 1 each time it is alive and 0 otherwise. To promote variable behavior in this agent as well, we implemented an $\epsilon$-greedy action selection (Supplemental Sec. E.4), where $\epsilon$ was chosen to match the average lifetime of the MOP agent (Supplemental Fig. E.2b). The behavior of the R agent was strikingly less variable than that of the MOP agent, spending more time close to the food source (Fig. 3b, R agent). Most importantly, while the MOP agent performs an almost equal number of clock and counterclockwise rotations, the R agent strongly prefers the clockwise rotations, reaching 90% of all observed rotations (Supplementary Movie 3, R-agent; Fig. 3c, orange line). This shows that the R agent mostly exploits only one strategy to survive and displays a smaller behavioral repertoire than the MOP agent.

## Dancing in an entropy-seeking cartpole

In the previous examples, complex behaviors emerge as a consequence of the presence of obstacles, predators, and limited food sources, but the actual dynamics of the agents are very coarse-grained. Here, we considered a system with physically realistic dynamics, the balancing cartpole[83,84], composed of a moving cart with an attached pole free to rotate (Fig. 4a). The cartpole is assumed to reach an absorbing state when either it hits a border, or when the pole angle exceeds 36°. Thus, we consider a broad range of angles that makes the agents reach a larger state space than in standard settings[85]. We discretized the state space and used a linear interpolation to solve for the optimal value function in Eq. (4), and to implement the optimal policy in Eq. (6) (Supplemental Sec. E.5). The MOP agent widely occupies the horizontal position, and more strikingly it produces a wide variety of pole angles, constantly swinging sideways as if it were dancing (Supplementary Movie 4, MOP agent; Fig. 4b, c).

We compared the behavior of an MOP agent with that of an R agent that receives a reward of 1 for being alive and 0 otherwise. The R agent gets this reward regardless of the pole angle and cart position within the allowed broad ranges, so that behaviors of the MOP and R agents can be better compared without explicitly favoring in any of them any specific behavior, such as the upright pole position. As expected, the R agent maintains the pole close to the balanced position throughout most of a long episode (Fig. 4b, bottom), because it is the furthest to the absorbing states and thus the safest. Therefore, the R agent produces very little behavioral variability (Fig. 4c, right panel) and no movement that could be dubbed "dancing" (Supplementary Movie 4, R agent). Although both MOP and R agents use a similar strategy which keeps the pole pointing towards the center for substantial amounts of time (Fig. 4c, positive angles correlate with positive positions in both panels), the behavior of the R agent is qualitatively

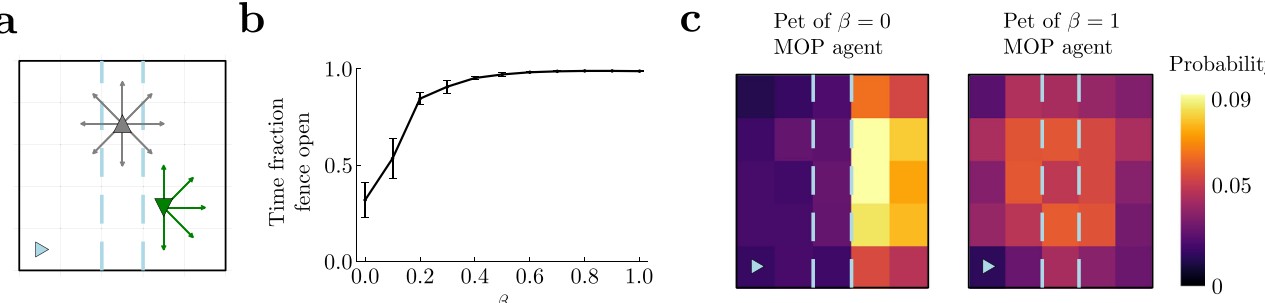

**Fig. 5 | Modeling altruism through an optimal tradeoff between own action entropy and other's state entropy. a** An agent (gray up triangle) has access to nine movement actions (gray arrows and doing nothing) and opens or closes a fence (dashed blue lines). This fence does not affect its movements. A pet (green, down triangle) has access to the same actions and chooses one randomly at each timestep, but it is constrained by the fence when closed. Pet location is part of the state of the agent. **b** As $\beta$ in Eq. (2) is increased, the agent tends to leave the fence open for a larger fraction of time. This helps its pet reach other parts of the arena. Error bars correspond to s.e.m. **c** Occupation heatmaps for 2000 timestep-episodes for $\beta = 0$ (left) and $\beta = 1$ (right). In all cases $\alpha = 1$.

different and is best described as a bang-bang sort of control for which the angle is kept very close to zero while the cart is allowed to travel and oscillate around the origin, which is more apparent in the actual paths of the agent (see trajectories in phase space in Supplementary Movie 5). We also find that the R agent does not display much variability in state space even after using an $\epsilon$-greedy action selection (Supplemental Fig. E.3, Supplementary Movie 6), with $\epsilon$ chosen to match average lifetimes between agents (Supplemental Fig. E.2c). This result showcases that the MOP agent exhibits the most appropriate sort of variability for a given average lifetime.

We finally introduced a slight variation to the environment, where the right half of the arena has stochastic state transitions, to showcase the ability of MOP to model risk-sensitive agents. Here, when agents choose an action (force) to be executed, a state transition in the desired direction occurs with probability $1 - \eta$, and a transition corresponding to zero force occurs with probability $\eta$ (Fig. 4d). Therefore, a MOP agent that seeks state entropy ($\beta > 0$) will show a preference for the right side, where there is in principle higher state entropy resulting from the stochastic transitions over more successor states than on the left side. Indeed, we find that MOP agents spend more time on the right side as $\beta$ increases, regardless of the probability $\eta$ (Fig. 4e). For fixed $\gamma$, spending more time on the right side can bring the life expectancy to decrease significantly depending on $\beta$ and $\eta$ (Supplemental Fig. E.2d, e). Interestingly, for $\beta > 0$ there is an optimal value of the noise $\eta$ that maximizes the fraction of time spent on the right side (Fig. 4f), which is a form of stochastic resonance. Therefore, for different $\beta$, qualitatively different behaviors emerge as a function of the noise level $\eta$.

### MOP agents can also seek entropy of others
Next, we considered an example where an agent seeks to occupy path space, which includes another agent's location as well as its own. The agent can freely move (Fig. 5a; gray triangle) and open or close a fence by pressing a lever in a corner (blue triangle). The pet of the agent (green triangle) can freely move if the fence is open, but when the fence is closed the pet is confined to move in the region where it is currently located. The pet moves randomly at each step, but its available actions are restricted by its available space (Supplemental Sec. E.6).

To maximize action-state path entropy, the agent ought to tradeoff the state entropy resulting from letting the pet free with the action entropy resulting from using the open-close action when visiting the lever location. The optimal tradeoff depends on the relative strength of action and state entropies. In fact, when state entropy weighs as much as action entropy ($\alpha = \beta = 1$), the fraction of time that the agent leaves the fence open is close to 1 (rightmost point in Fig. 5b) so that the pet is free to move (Fig. 5c, right panel; $\beta = 1$ MOP agent). However, when the state entropy has zero weight ($\alpha = 1, \beta = 0$), the fraction of time that the fence remains open is close to 0.5 (leftmost point in Fig. 5b), and the pet remains confined to the right side for most of the time (Fig. 5c, left panel; $\beta = 0$ MOP agent), the region where it was initially placed. As a function of $\beta$ the fraction of time the fence is open increases. Therefore, the agent gives more freedom to its pet, as measured by the pet's state entropy, by curtailing its own action freedom, as measured by action entropy, thus becoming more "altruistic".

### MOP compared to other reward-free approaches
One important question is how MOP compares to other reward-free, motivation-driven theories of behavior. Here we focus on two popular approaches: empowerment and the FEP. In empowerment (MPOW)[20,73–75] agents maximize the mutual information between $n$-step actions and the successor states resulting from them[20,86], a measure of their capability to perform diverse courses of actions with predictable consequences. MPOW formulates behavior as greedy maximization of empowerment[20,73], such that agents move to accessible states with the largest empowerment (maximal mutual information), and stay there with high probability.

We applied MPOW to the grid world and cartpole environments (Fig. 6). In the gridworld, MPOW agents (5-step MPOW, see Supplemental Sec. G.1) prefer states from where they can reach many distinct states, such as the middle of a room. However, due to energetic constraints, they also gravitate towards the food source when energy is low, and they alternate between these two locations ad nauseam (Fig. 6a, middle; Supplementary Movie 7). In the cartpole, MPOW agents (3-step MPOW[73], see Supplemental Sec. G.1) favor the upright position because, being an unstable fixed point, it is the state with highest empowerment, as previously reported[73,87]. Given the unstable equilibrium, the MPOW agent gets close to it but needs to continuously adjust its actions when greedily maximizing empowerment (Fig. 6b, middle; Supplementary Movie 8). The paths traversed by MPOW agents in state space are highly predictable, and they are similar to the ones of the R agent (see Fig. 4c). The only source of stochasticity comes from the algorithm, which approximately calculates empowerment, and thus a more precise estimation of empowerment leads to even less variability.

In the FEP, agents seek to minimize the negative log probability, called surprise, of a subset of desired states via the minimization of an upper bound, called free energy. This minimization reduces behavioral richness by making a set of desired (homeostatic) states highly likely[76,77], rendering this approach almost opposite to MOP. In a recent MDP formalization, FEP agents aim to minimize the (future) expected free energy (EFE)[88], which equals the future cumulative KL divergence between the probability of states and the desired (target) probability of those states (see Supplemental Sec. G.2 for details). Even though this

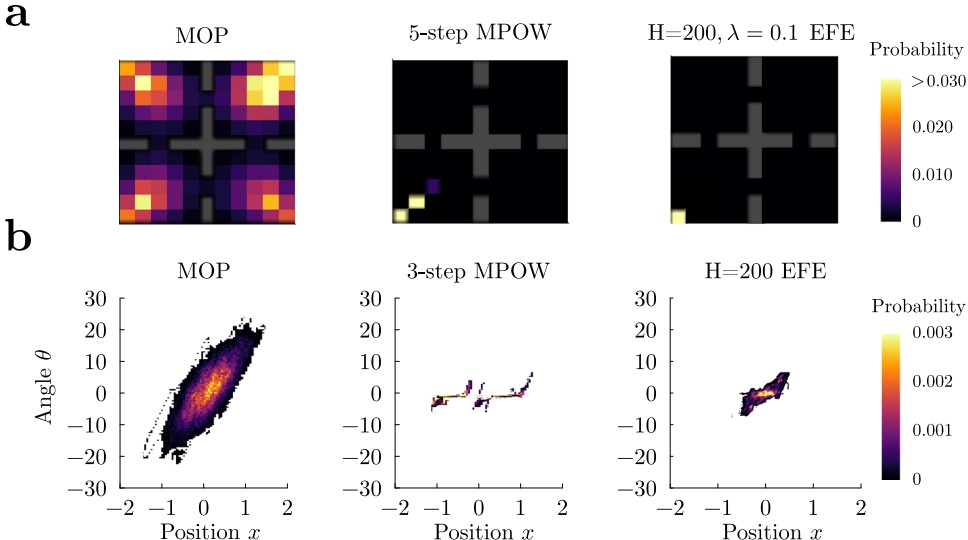

**Fig. 6 | Empowerment (MPOW) and Free Energy Principle (EFE) lack robust occupation of action-states. a** In the grid-world environment, MPOW and expected free energy (EFE) only visit a restricted portion of the arena. Initial position was the center of a room $(x, y) = (3, 3)$. **b** In the cartpole environment, both MPOW and EFE shy away from large angles, producing a limited repertoire of predictable behaviors.

objective contains the standard exploration entropy term on state transitions[77,89], we prove that the optimal policy is deterministic (see Supplemental Sec. G.2).

As a consequence, we find that in both the gridworld and cartpole environments, the behavior of the EFE agent (receding horizon $H = 200$) is much less variable than the MOP agent in general (Fig. 6a, right panel for the gridworld, Supplementary Movie 7; and b, right panel, for the cartpole, Supplementary Movie 8). The only source of action variability in the EFE agent is due to the degeneracy of the EFE, and thus behavior collapses to a deterministic policy as soon as the target distribution is not perfectly uniform (see Supplemental Sec. G.2.2 for details). We finally prove that under discounted infinite horizon and assuming a deterministic environment, the EFE agent is equivalent to a classical reward maximizer agent with reward $R = 1$ for all non-absorbing states and $R = 0$ for the absorbing states (Supplemental Sec. G.2). In conclusion, MOP generates much more variable behaviors than MPOW and FEP.

## MOP in continuous and large action-state spaces

The examples so far can be solved exactly with numerical methods, without relying on function approximation of the value function or the policy, which could obscure the richness of the resulting behaviors. However, one important question is whether our approach scales up to large continuous action-state spaces where no exact solutions are available. To show that MOP generates rich behaviors even in high-dimensional agents, we simulated a quadruped via MuJoCo[90] from Gymnasium[91] without imposing any explicit fine-tuned reward function (Fig. 7a). The only externally imposed conditions are the absorbing states, which are reached when either the agent falls (given by the torso touching the ground), or the torso reaches a maximum height[91].

We first trained the MOP agent by approximating the state-value function, Eq. (5), using the soft-actor critic (SAC) architecture[49] with zero rewards, which corresponds to the case $\alpha = 1$ and $\beta = 0$. The MOP agent learns to stabilize itself and walk around, sometimes jumping, spinning, and moving up and down the legs, without any instructions to do so (Supplementary Movie 9). The MOP agent exhibits variable and long excursions over state space (Fig. 7b, c blue) and displays a broad distribution of speeds (Fig. 7d, blue). We compared the MOP agent with an R agent that obtains a reward of $R = 1$ whenever it is alive and $R = 0$ when it reaches an absorbing state. As before, we add variability to the R agent with an $\epsilon$-greedy action selection, adjusting $\epsilon$

so that the average lifetime of the R agent matched that of the MOP agent (Supplemental Fig. E.4a). In contrast to the MOP agent, the R agents exhibit much shorter excursions (Fig. 7b, c yellow) and a velocity distribution that peaks around zero, indicating prolonged periods spent with no translational movement (Fig. 7d, yellow). When visually compared, the behavior for MOP and R agents shows stark differences (Supplementary Movie 9).

While the MOP agent elicits variable behaviors, it is also capable of generating deterministic, goal-directed behaviors when needed. To show this, we added a food source in the arena and extended the state of the agent with its internal energy. Now the agent can also die of starvation when the internal energy hits zero (Fig. 7e). As expected, when the initial location of the MOP quadruped is far from the food source, it directly moves to the food source to avoid dying from starvation (Fig. 7f). After the food source is reached for the first time, the MOP quadruped generates random excursions away from the food source. During these two phases, the agent displays very different speed distributions (Fig. 7g), showing also quantitative differences in the way it moves (see a comparison with the R agent in Supplemental Fig. E.4, and Supplementary Movie 10).

Finally, we modified the environment by adding state transition noise of various magnitudes in one half of the arena ($x > 0$), while the other half remained deterministic. We find that the agent's behavior is modulated by $\beta$, which controls the preference of state transition entropy (see details in Supplemental Sec. E.7). As expected, for fixed $\alpha$ and positive noise magnitude, MOP agents show increasing preference toward the noisy side as $\beta$ increases (Supplemental Fig. E.5). However, as noise magnitude increases, and for fixed $\beta$, MOP agents tend to avoid the noisy side to prevent them from falling. This shows that MOP agents can exhibit approach and avoidance behaviors depending on the environment's stochasticity and their $\beta$ hyperparameter.

## Discussion

Often, the success of agents in nature is not measured by the amount of reward obtained, but by their ability to expand in state space and perform complex behaviors. Here we have proposed that a major goal of intelligence is to "occupy path space". Extrinsic rewards are thus the means to move and occupy action-state path space, not the goal of behavior. In an MDP setting, we have shown that the intuitive notion of path occupancy is captured by future action-state path entropy, and we have proposed that behavior is driven by the maximization of this

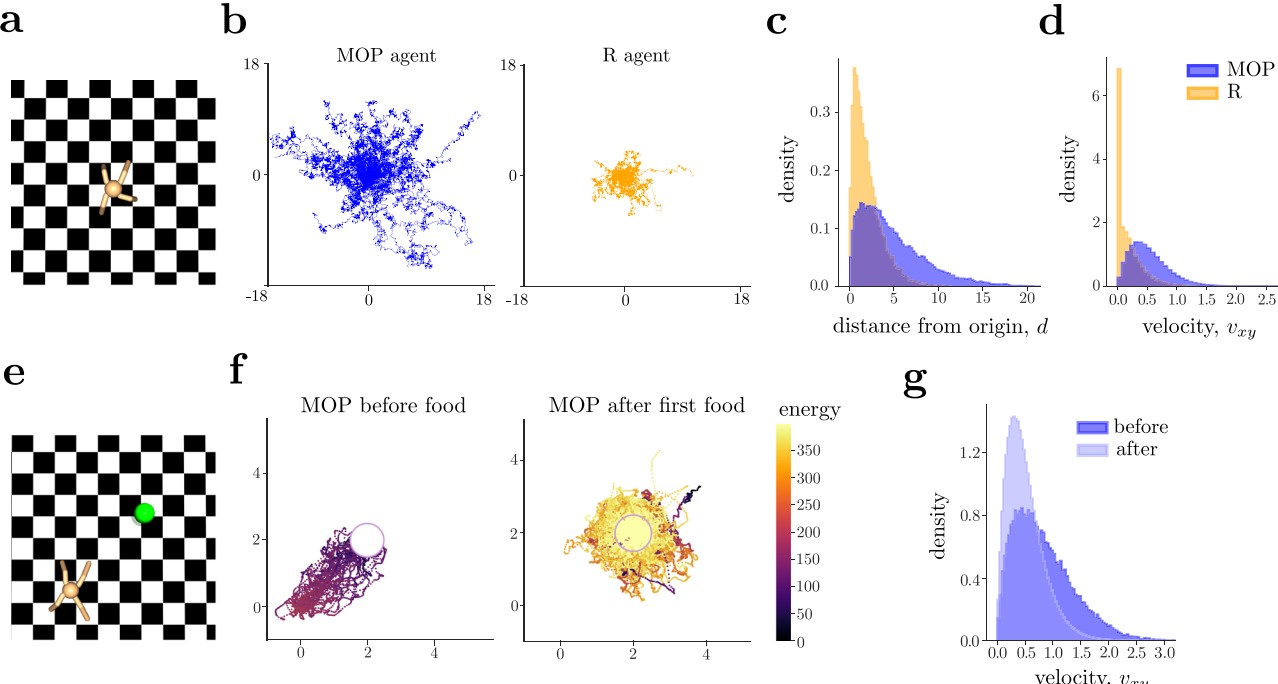

**Fig. 7 | MOP in high-dimensional states generates variable and goal-directed behaviors. a** The quadruped environment, adapted from Gymnasium, serves as the testing environment. The $x, y$ dimensions are unbounded. **b** Trajectories of the center of mass of the torso of the MOP (left panel) and R (right panel) agents. MOP occupies more space for approximately the same survival time (see Supplemental Fig. E.4a). Distribution of the planar distance $d$ from the origin (**c**) and planar speed $v_{xy}$ (**d**) for MOP (blue) and R (yellow) agents. **e** In a new environment, a food source

(green ball) is available so that the MOP agent can replenish its internal energy to avoid starvation. **f** Trajectories of the MOP agent before (left) and after (right) getting to the food source. Colormap defined by the energy level of the agent. **g** Distribution of the planar speed showcasing changes before (dark blue) and after (light blue) the MOP agent reaches the food source for the first time. Distributions computed only on the tests where the quadruped finds the food source.

intrinsic goal—the MOP. We have solved the associated Bellman equation and provided a convergent iterative map to determine the optimal policy. In several discrete and continuous state examples, we have shown that MOP, along with the agent's constraints and dynamics, leads to complex behaviors that are not observed in other simple reward-maximizing agents. Quick filling of physical space by a moving agent, hide-and-seek behavior and variable escaping routes in a predator-prey example, dancing in a realistic cartpole dynamical system, altruistic behavior in an agent-and-pet duo and successful, vigorous movement in a high-dimensional quadruped are all behaviors that strike as being playful, curiosity-driven and energetic.

To the human eye, these behaviors look genuinely goal-directed, like approaching the food source when the energy level is low or escaping from the cat when it gets close to the mouse (see Figs. 2c and 3d). Although MOP agents do not have any extrinsically designed goal, like eating or escaping, they generate these deterministic, goal-directed behaviors whenever necessary so that they can keep moving in the future and maximize future path action-state entropy (see Supplemental Sec. H). These results show that the presence of internal states (e.g., energy) and absorbing states (e.g., having zero energy or being eaten) are critical for generating interesting behaviors, as getting close to different types of absorbing states triggers qualitatively different behaviors. This capability of adapting variability depending on internal states has been overlooked in the literature and is essential to obtaining the goal-directed behaviors we have shown here.

While all reward-maximizing principles of behavior predict that learning reduces variability and it collapses to zero once learning is deemed to have finished (i.e., there always exists an optimal deterministic policy[92]), MOP predicts that behavioral stochasticity persists even after learning (i.e., optimal policies under MOP are

non-deterministic). This would imply that noise is promoted as long as it does not compromise the functioning of the agent, even after extensive experience with an environment. This prediction is supported by observations across different living organisms and at different levels of organization, including perception, behavior, neural circuits, and individual neurons. At the perceptual level, a paradigmatic example of the ubiquity of stochasticity is multistable perception, observed in humans[93] and other animals[94–97], a phenomenon consisting of a stochastic alternation between percepts that never stops, even though stimuli are simple and familiar. At the behavioral level, stochastic actions are observed in all kinds of living organisms. For example, foraging behavior in bacteria, plants, and animals is well-described by random walks and Lévy flights[30,98,99], which remain valid descriptions even in familiar environments. Further, mice are found to perform random choices from time to time in simple binary tasks even after extended periods of training[32]. Neuroanatomically, emerging evidence indicates that specific neural circuits play a crucial role in generating behavioral variability. A striking example is the avian basal ganglia–forebrain circuit, which modulates the variability of the birds' songs[100,101], even after learning. Indeed, adult male birds with learned motor skills exhibit a higher level of variability when they sing alone, compared to during courtship[102,103]. At the neural level, single-neuron responses across sensory and motor areas are highly variable, and this variability persists after long periods of stimulation and adaptation[104–106]. All this evidence points out that, at least in part, behavior and its associated neural circuits and components are stochastic, regardless of the state of learning.

Not only do organisms seem to generate variability, but they are also sensitive to the state-transition uncertainty in or about the environment, a phenomenon known as risk sensitivity. Risk-seeking

and risk-aversive behaviors are both found in humans[33,107], non-human primates[34,108], rodents, birds and other species[38]. While other works have looked at risk-sensitivity in RL, it is commonly associated to sensitivity to variations in extrinsic rewards[36,37]. In contrast, MOP can model risk-sensitivity by weighing the state-transition entropy with $\beta$. By adding stochastic regions to otherwise noiseless environments (four-room gridworld, cartpole, and quadruped), we showed that MOP can model risk-sensitive agents in the absence of extrinsic reward. For example, for the right combination of state entropy weight $\beta$, and lookahead horizon, controlled by $\gamma$, MOP agents could get stuck in a noisy TV, consistent with the observation that humans have a pre-ference for noisy TVs under particular conditions[109]. However, it can also capture the avoidance of noisy TVs for sufficiently long sighted agents (see Fig. 2e). These observations and results show that MOP could provide a unified account for the stochastic nature of behavior and its risk sensitivity, offering the opportunity to model such see-mingly disparate set of observations by modulating its action and state entropy contributions.

A set of algorithms related to MOP, known as empowerment, has also proposed using reward-free objectives as the goal of behavior[20,73,75]. In this approach, the mutual information between a sequence of actions and the final state is maximized. This makes empowerment agents prefer states where actions lead to large and predictable changes, such as unstable fixed points[73]. We have shown that one drawback is that empowered agents tend to remain close to those states without producing diverse behavioral repertoires (see Fig. 6b and Supplementary Movie 8), as it also happens in causal entropy approaches[110]. Another difference is that empowerment is not additive over paths because the mutual information of a path of actions with the path of states is not the sum of the per-step mutual information, and thus it cannot be formalized as a cumulative per-step objective (Supplemental Sec. I)[73,75,81,111], in contrast to action-state path entropy. We note, however, that an approximation to empowerment having the desired additive property could be obtained from our fra-mework by putting $\beta < 0$ in Eq. (2), such that more predictable state transitions are preferred. Similarly to empowerment, we have also shown that agents following the FEP[76,88] collapse behavior to deter-ministic policies in known environments (see Fig. 6b and Supple-mentary Movie 8). Other reward-free RL settings and pure exploration objectives have been proposed in the past[57,59,61,67,112–115], but this body of work typically investigates how to efficiently sample MDPs to con-struct near-optimal policies when reward functions are introduced in the exploitation phase. More importantly, this work differs from ours in that the goal-directedness that MOP displays entails behavioral variability at its core, even in known environments (see examples above). Finally, other overlapping reward-free approaches focus on the unsupervised discovery of skills, by encouraging diversity[26,116–118]. While the motivation is similar, they focus on skill-conditioned policies, whereas our work demonstrates that complex sequences of behaviors are possible working from the primitive actions of agents, although a possible future avenue for MOP is to apply it to temporally extended actions[119]. In addition, these works define tasks based on extrinsic rewards, whereas we have shown that internal state signals are suffi-cient to let agents define sub-tasks autonomously.

Our approach is conceptually different as well from hybrid approaches that combine extrinsic rewards with action entropy or KL regularization terms[46,47,50,52,120] for two main reasons. First, entropy-seeking behavior does not pursue any form of extrinsic reward max-imization. But most importantly, using KL-regularization using a default policy $\pi_0(a|s)$ in our framework would be self-defeating. This is because the absolute action entropy terms $\mathcal{H}(A|s)$ in the expected return in Eq. (2) favor visiting states where a large set of immediate and future action-states are accessible. In contrast, using relative action entropy (KL) precludes this effect by normalizing the number of accessible actions, as we have shown above. Additionally, minimizing

the KL divergence with a uniform default policy and without extrinsic rewards leads to an optimal policy that is uniform regardless of the presence of absorbing states, equivalent to a random walk, which shows that a pure KL objective does not lead to interesting behaviors (Supplemental Sec. F, Supplemental Fig. F.6). The idea of having a variable number of actions that depend on the state is consistent with the concept of affordance[121]. While we do not address the question of how agents get the information about the available actions, an option would be to use the notion of affordances as actions[122]. Secondly, while previous work has studied the performance benefits of either action[49], state[51,57] or equally weighted action-state[53,123] steady-state entropies, our work proposes mixing them arbitrarily through path entropy, leading to a more general theory without any loss in mathematical tractability[56].

We have also shown that MOP is scalable to high-dimensional problems and when the state-transition matrix is unknown, using the SAC architecture[124] to approximate the optimal policy prescribed by MOP. Nevertheless, several steps remain to have a more complete MOP theory with learning. Previous related attempts have introduced Z-learning[46,79] and G-learning[125] using off-policy methods, so our results could be extended to learning following similar lines. Other possibi-lities are using transition estimators using counts or pseudo-counts[69], or hashing[70], for the learning of the transition matrices. One potential advantage of our framework is that, as entropy-seeking behavior obviates extrinsic rewards, those rewards do not need to be learned and optimized, and thus the learning problem reduces to transition matrices learning. In addition, modeling and injecting prior informa-tion could be particularly simple in our setting in view that intrinsic entropy rewards can be easily bounded before the learning process if action space is known. Therefore, initializing the state-value function to the lower or upper bounds of the action-state path entropy could naturally model pessimism or optimism during learning, respectively.

All in all, we have introduced MOP as a novel theory of behavior, which promises new ways of understanding goal-directedness without reward maximization, and that can be applied to artificial agents to discover by themselves ways of surviving and occupying action-state space.

## Methods
### Entropy measures the occupancy of action-state paths
We consider a time-homogeneous MDP with finite state set $\mathcal{S}$ and finite action set $\mathcal{A}(s)$ for every state $s \in \mathcal{S}$. Henceforth, the action-state $x_j = (a_j, s_j)$ is any joint pair of one available action $a_j$ and one possible successor state $s_j$ that results from making that action under policy $\pi \equiv \{\pi(a|s)\}$ from the action-state $x_i = (a_i, s_i)$. By assumption, the avail-ability of action $a_j$ depends on the previous state $s_i$ alone, not on $a_i$. Thus, the transition probability from $x_i$ to $x_j$ in one-time step is $p_{ij} = \pi(a_j|s_i)p(s_j|s_i, a_j)$, where $p(s_j|s_i, a_i)$ is the conditional probability of transitioning from state $s_i$ to $s_j$ given that action $a_j$ is performed. Although there is no dependence of the previous action $a_i$ on this transition probability, it is notationally convenient to define transitions between action-states.

We conceive of rational agents as maximizing future action-state path occupancy. Any measure of occupancy should obey the intuitive Conditions $1$–$4$ listed below.

*Intuitive Conditions for a measure of action-state path occupancy:*
1. Occupancy gain of action-state $x_j$ from $x_i$ is a function of the transition probability $p_{ij}$, $C(p_{ij})$
2. Performing a low probability transition leads to a higher occu-pancy gain than performing a high probability transition, that is, $C(p_{ij})$ decreases with $p_{ij}$
3. The first order derivative $C'(p_{ij})$ is continuous for $p_{ij} \in (0, 1)$
4. (Definition: the action-state occupancy of a one-step path from action-state $x_i$ is the expectation over occupancy gains of the immediate successor action-states, $C_i^{(1)} \equiv \sum_j p_{ij}C(p_{ij})$). The action-

state occupancy of a two-step path is additive, $C_i^{(2)} \equiv \sum_{jk} p_{ij} p_{jk} C(p_{ij} p_{jk}) = C_i^{(1)} + \sum_j p_{ij} C_j^{(1)}$ for any choice of the $p_{ij}$ and initial $x_i$

Condition 1 simply states that occupancy gain from an initial action-state is defined over the transition probabilities to successor action-states in a sample space. Condition 2 implies that performing a low-probability transition leads to a higher occupancy of the successor states than performing a high-probability transition. This is because performing a rare transition allows the agent to occupy a space that was left initially unoccupied. Condition 3 imposes smoothness of the measure.

In Condition 4 we have defined the occupancy of the successor action-states (one-step paths) in the Markov chain as the expected occupancy gain. Condition 4 is the central property, and it imposes that the occupancy of action-states paths with two steps can be broken down into a sum of the occupancies of action-states at each time step. Note that the action-state path occupancy can be written as

$$C_i^{(2)} \equiv \sum_{jk} p_{ij} p_{jk} C(p_{ij} p_{jk}) = \sum_j p_{ij} C(p_{ij})$$
$$+ \sum_{jk} p_{ij} p_{jk} C(p_{jk}) = \sum_{jk} p_{ij} p_{jk} \Big( C(p_{ij}) + C(p_{jk}) \Big),$$

which imposes a strong condition on the function $C(p)$. Note also that the sum $\sum_{jk} p_{ij} p_{jk} C(p_{ij} p_{jk})$ extends the notion of action-state to a path of two consecutive action-states, each path having probability $p_{ij} p_{jk}$ due to the (time-homogeneous) Markov property. The last equality is an identity. While here we consider paths of length equal to 2, in the Supplementary Information we show that there is no difference in imposing additivity to paths of any fixed or random length (Corollary 2). The additivity of occupancy over paths is equivalent to time homogeneity, that is, all times count the same.

**Theorem 1.** $C(p) = -k \ln p$ with $k > 0$ is the only function that satisfies Conditions 1–4.

**Corollary 1.** The entropy $C_i^{(1)} = -k \sum_j p_{ij} \ln p_{ij}$ is the only measure of action-state occupancy of successor action-states $x_j$ from $x_i$ with transition probabilities $p_{ij}$ consistent with Conditions 1–4.

See Supplementary Information for the proof. We have found that entropy is the measure of occupancy. Shannon's famous derivation of entropy as a measure of information[78] uses similar elements, but some differences are worthy to be mentioned. First, our proof uses the notion of additivity of occupancy on MDPs of length two (our Condition 4), while Shannon's notion of additivity uses sequences of random variables of arbitrary length (his Condition 3), and therefore his condition is in a sense stronger than ours. Second, our proof enforces continuous derivative of the measure, while Shannon enforces continuity of the measure, rendering our Condition 3 stronger. Finally, we enforce a specific form of the measure as an average over occupancy gains (our Condition 4 again), because it intuitively captures the notion of occupancy, while Shannon does not enforce this structure in his information measure.

## Critical values and policies

**Theorem 2.** The critical values $V^c(s)$ of the expected returns $V_\pi(s)$ in equation (3) with respect to the policy probabilities $\pi = \{\pi(a|s): a \in A(s), s \in S\}$ obey

$$V^c(s) = \alpha \ln Z(s) = \alpha \ln \left[ \sum_{a \in \mathcal{A}(s)} \exp \left( \alpha^{-1} \beta \mathcal{H}(S'|s,a) + \alpha^{-1} \gamma \sum_{s'} p(s'|s,a) V^c(s') \right) \right]$$
(8)

where $\mathcal{H}(S'|s,a) = -\sum_{s'} p(s'|s,a) \ln p(s'|s,a)$ is the entropy of the successors of $s$ after performing action $a$, and $Z(s)$ is the partition function.

The critical points (critical policies) are

$$\pi^c(a|s) = \frac{1}{Z(s)} \exp \left( \alpha^{-1} \beta \mathcal{H}(S'|s,a) + \alpha^{-1} \gamma \sum_{s'} p(s'|s,a) V^c(s') \right), \quad (9)$$

one per critical value, where the partition function $Z(s)$ is the normalization constant.

Note that we simultaneously optimize $|S|$ expected returns, one per state $s$, each with respect to the set of probabilities $\pi = \{\pi(a|s): a \in A(s), s \in S\}$.

**Proof.** We first note that the expected return in Eq. (2) is continuous and has continuous derivatives with respect to the policy except at the boundaries (i.e., $\pi(a|s) = 0$ for some action-state $(a, s)$). Choosing a state $s$, we first take partial derivatives with respect to $\pi(a|s)$ for each $a \in \mathcal{A}(s)$ in both sides of Eq. (3) and then evaluate them at a critical point $\pi^c$ to obtain the condition

$$\lambda(s,s) = \sum_{s'} p(s'|s,a) \Big( -\ln(\pi^c(a|s)^\alpha p^\beta(s'|s,a)) + \gamma V^c(s') \Big)$$
$$- \alpha + \gamma \sum_{b,s'} \pi^c(b|s) p(s'|s,b) \lambda(s',s)$$
$$= -\alpha \ln \pi^c(a|s) - \beta \sum_{s'} p(s'|s,a) \ln p(s'|s,a) - \alpha$$
$$+ \gamma \sum_{s'} p(s'|s,a) V^c(s') + \gamma \sum_{b,s'} \pi^c(b|s) p(s'|s,b) \lambda(s',s),$$
(10)

where we have defined the partial derivative at the critical point $\frac{\partial V_\pi(s')}{\partial \pi(a|s)}|_{\pi^c} \equiv \lambda(s',s)$ and used the fact that this partial derivative should be action-independent. To understand this, note that the critical policy should lie in the simplex $\sum_a \pi(a|s) = 1$, $\pi(a|s) \geq 0$, and therefore the gradient of $V_\pi(s')$ with respect to the $\pi(a|s)$ at the critical policy should be along the normal to the constraint surface, i.e., the diagonal direction (hence, action-independent), or be zero. Indeed, the action-independence of the $\lambda(s',s)$ also results from interpreting them as Lagrange multipliers: $\lambda(s',s)$ is the Lagrange multiplier corresponding to the state-value function at $s'$, $V_\pi(s')$, associated to the constraint $\sum_a \pi(a|s) = 1$, $\pi(a|s) \geq 0$, defining the simplex where the probabilities $\{\pi(a|s): a \in A(s)\}$ lie.

Noticing that the last term of Eq. (10) does not depend on $a$, we can solve for the critical policy $\pi^c(a|s)$ to obtain Eq. (9). Eq. (9) implicitly relates the critical policy with the critical value of the expected returns from each state $s$. Inserting the critical policy, Eq. (9), into Eq. (3), we get Eq. (8), which is an implicit non-linear system of equations exclusively depending on the critical values.

It is easy to verify that the partial derivatives of $V_\pi(s)$ in Eq. (3) with respect to $\pi(a'|s')$ for $s \neq s'$ are

$$\lambda(s,s') = \gamma \sum_{s''} p(s''|s) \lambda(s'',s'),$$

and thus they provide no additional constraint on the critical policy. $\square$

We further prove in the Supplementary Information (Theorems 3 and 4) that the critical point is unique, corresponds to the optimal policy, and that the optimal state-value function can efficiently be obtained from the iterative mapping in Eq. (7).

## Definition of agents
**MOP agent.** In all the experiments presented, we introduce the MOP agent. The objective function that this agent maximizes, in general, is Eq. (2). As described in the manuscript, the $\alpha$ and $\beta$ parameters control the weights of action and next-state entropies to the objective function, respectively, but only their ratio $\beta/\alpha$ is important. Unless

indicated otherwise, we always use $\alpha = 1, \beta = 0$ for the experiments. It is important to note that if the environment is deterministic, then the next-state entropy $\mathcal{H}(S'|s,a) = -\sum_{s'} p(s'|s,a) \ln p(s'|s,a) = 0$, and therefore $\beta$ does not change the optimal policy, Eq. (6).

The number of actions and state transitions available can depend on the $s$. In particular, we consider absorbing states $s^+$, where only the "stay" action available. This construction leads to states with zero action and state transition entropies, and thus absorbing states have zero state-value $V_\pi(s^+) = 0$. While a reward function (see next section) tells an agent what needs to be accomplished, absorbing states tell a MOP agent what not to do. Thus, to occupy action-state path space, a MOP agent fabricates by itself a variety of goals and subgoals that are solely constrained by the presence of absorbing states and the environment-agent dynamics.

We have implemented the iterative map, Eq. (7), to solve for the optimal value, using $z_i^{(0)} = 1$ for all $i$ as initial condition. Theorem (3) ensures that this iterative map finds a unique optimal value regardless of the initial condition in the first orthant. To determine a degree of convergence, we compute the supremum norm between iterations,

$$\delta = \max_i |V_i^{(n+1)} - V_i^{(n)}|,$$

where $V_i = \frac{\alpha}{\gamma} \log(z_i)$, and the iterative map stops when $\delta < 10^{-3}$.

**R agent.** We also introduce a reward-maximizing agent in the usual RL sense. In this case, the reward is $r = 1$ for living and $r = 0$ when dying. In other words, this agent maximizes life expectancy. Additionally, to emphasize the typical reward-seeking behavior and avoid degenerate cases induced by the tasks, we introduced a small reward for the Four-room grid world (see Supplementary Information for details of the environments). In all other aspects, the modeling of the R agent is identical to the MOP agent. To allow for reward-maximizing agents to display some stochasticity, we used an $\epsilon$-greedy policy, the best in the family of $\epsilon$-soft policies[14]. At any given state, a random admissible action is chosen with probability $\epsilon$, and the action that maximizes the value is chosen with probability $1 - \epsilon$. Given that the world models $p(s'|s,a)$ are known and the environments are static, this $\epsilon$-greedy policy does not serve the purpose of exploration (in the sense of learning), but only to inject behavioral variability. Therefore, we construct an agent with state-independent variability, whose value function satisfies the optimality Bellman equation for this $\epsilon$-greedy policy,

$$V_\epsilon(s) = (1-\epsilon)\max_a \sum_{s'} p(s'|s,a)\left(r + \gamma V_\epsilon(s')\right) + \frac{\epsilon}{|\mathcal{A}(s)|} \sum_{a,s'} p(s'|s,a)\left(r + \gamma V_\epsilon(s')\right),$$

(11)

where $|\mathcal{A}(s)|$ is the number of admissible actions at state $s$. To solve for the optimal value in this Bellman equation, we perform value iteration[14]. The $\epsilon$-greedy policy for the R agent is therefore given by

$$\pi(a|s) = \begin{cases} 1 - \epsilon + \frac{\epsilon}{|\mathcal{A}(s)|}, & \text{if } a = \arg\max_{a'} \sum_{s'} p(s'|s,a')\left(r + \gamma V_\epsilon(s')\right) \\ \frac{\epsilon}{|\mathcal{A}(s)|}, & \text{otherwise} \end{cases}$$

where ties in arg max are broken randomly. Note that if $\epsilon = 0$, we obtain the usual greedy optimal policy that maximizes reward. The parameter $\epsilon$ is typically chosen so that average lifetimes for both R and MOP agents are matched.

### Reporting summary
Further information on research design is available in the Nature Portfolio Reporting Summary linked to this article.

## Data availability
The data generated in this study have been generated through a custom code[126]. The specific quadruped data analyzed in this study are available under request, although an equivalent dataset can be generated with the code provided.

## Code availability
The code to generate the results and various figures is available as Python and Julia code along with guided notebooks to reproduce the figures[126].

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

## Acknowledgements

This work is supported by the Howard Hughes Medical Institute (HHMI, ref 55008742), ICREA Academia 2022 and MINECO (Spain; BFU2017-85936-P) to R.M.-B., MINECO/ESF (Spain; PRE2018-084757) to J.R.-R, and AGAUR-FI ajuts from Generalitat de Catalunya/ESF (2024 FI-B3 00020) to C.M and (2023 FI-1 00245) to Y.H.

## Author contributions

J.R.-R. and R.M.-B. conceived the presented idea. J.R.-R., D.G., and R.M.B. developed the theory. J.R.-R., D.G., C.M., and Y.H. performed the computations and analyzed the data. All authors discussed the results and contributed to the final manuscript.

## Competing interests

The authors declare no competing interests.
