## [Peer Review File · Nature Communications]

REVIEWER COMMENTS

Reviewer #1 (Remarks to the Author):

The authors propose state-action entropy maximization as a fundamental principle underlying the learning of behaviors. It consists in maximizing the. They exhibit on toy environments that some interesting behaviors can emerge. The structure of the paper is overall easy to follow, even though some introduction sentences could help a reader in the subsections of the method section.

The significance and originality of the paper suffers from unclear conceptual and experimental differences with other methods and the lack of a clear and sound evaluation protocol, as explained below.

Major points

Unclear positioning

The paper proposes a behavior principle but lacks some discussion about different behavior principles, like the Free-energy principle [2] or compression maximization [3]. Intrinsic motivations in RL (45-53) are indeed evaluated on RL benchmarks, but they mostly aim to assess their exploration ability. The final objective is not to solve an arbitrary Atari game. In addition, the trade-off between intrinsic and extrinsic rewards is often ruled by one single hyper-parameter which could be set to 0. Therefore, it is bad faith to state that the behavioral variability ceases after learning as a general rule. With this in mind, numerous works can be described as maximizing state entropy (see Section 6 of [4]) and some of these methods are/can easily be associated with action-entropy maximization RL algorithms like SAC [5]. So, there already exist methods that aim to simultaneously maximize both state entropy and action entropy: what is the fundamental difference with theirs ?

Lack of comparison to other intrinsic motivations

One of the main experimental contribution of the paper is to show the emergence of particular behaviors. But it remains unclear whether a more common state entropy maximization would end up

with similar outcomes. Appendix F is a first step in this direction and it should be somehow integrated in the main paper, but similar variants of the loss should also be tested in the other environments. It would also give more insights about the relative roles of the action and state entropy terms. Another interesting baseline could be an intrinsic reward of 1, since the positivity of the reward may induce keep-alive behaviors.

How are goal-directed behaviors defined ?

The authors claim that the agent learns goal-directed behaviors (abstract and conclusion) and it is one of the main results. I do not understand the difference between the goal-directed behaviors (e.g. food seeking) and a simple policy. In the gridworld arena of Section 3.1, we are visually biased to consider the different cells as the only state space, but since internal energy is part of the state space, what is so fantastic about the policy changing according to the energy level ?

Minor points:

Limited environments:

I think the paper deserves a discussion about the complexity of the environments, what happens if they progressively add more pets in 3.4 ? How could the method work with continuous states/action spaces or with larger number of states ?

Unclear equation:

I do not understand how they can go from Equation 1 to Equation 2. They replaced $R(\tau)$ by equation 1, but the modified term is not the same. It looks like the expectation term should no longer be there. Please could the authors clarify that ?

External vs extrinsic rewards

The authors explain they abandon external rewards in page 2, and then they explain they define an external reward function. It may come from a confusion between external rewards and extrinsic rewards. This confusion holds all along the paper and should be clarified. If they consider different definitions from the existing literature [1], it should be explicitly stated and argued for.

Bug in prey-predator ?

In video 3, it happens that the cat touches the mouse, but she does not die (e.g. 6s) despite what is written in the paper. Why ?

Typos:

Page 1 - introduction: Further, setting a reward function by design as the goal of artificial agents is more often than not arbitrary

Page 2 - introduction: action-sate

Page 5 - results: they confuse Figure 1 and Figure 2.

Others:

Please, send .mp4 video files.

References

- [1] Oudeyer, P. Y., & Kaplan, F. (2009). What is intrinsic motivation? A typology of computational approaches. *Frontiers in neurorobotics*, 6.
- [2] Friston, K., Kilner, J., & Harrison, L. (2006). A free energy principle for the brain. *Journal of physiology-Paris*, 100(1-3), 70-87.
- [3] Schmidhuber, J. (2009). Driven by compression progress: A simple principle explains essential aspects of subjective beauty, novelty, surprise, interestingness, attention, curiosity, creativity, art, science, music, jokes. In *Anticipatory Behavior in Adaptive Learning Systems: From Psychological Theories to Artificial Cognitive Systems 4* (pp. 48-76). Springer Berlin Heidelberg.
- [4] Aubret, A., Matignon, L., & Hassas, S. (2023). An information-theoretic perspective on intrinsic motivation in reinforcement learning: a survey. *Entropy*, 25(2), 327.
- [5] Haarnoja, T., Zhou, A., Abbeel, P., & Levine, S. (2018, July). Soft actor-critic: Off-policy maximum entropy deep reinforcement learning with a stochastic actor. In *International conference on machine learning* (pp. 1861-1870). PMLR.

Reviewer #2 (Remarks to the Author):

The paper presents an MDP framework with known transition dynamics where instead of considering the maximization of an external reward signal it is considered the maximization of action entropy and future state entropy. This objective is proposed as “the sole goal of intelligent behavior” and as a proxy for obtaining curiosity-driven behavior. Theoretical justification is provided for this objective, and experiments on small domains are conducted showing the applicability of the methodology.

What are the noteworthy results?

There is currently a hot debate on what are rewards in RL and where do they come from [1]. The proposal of not using rewards to obtain seemingly intelligent policies, as proposed by this work and others[2] is interesting in its own right.

The manuscript is well written and easy to understand. There has been a significant effort to clearly explain the experiments and the mathematical statements.

The present manuscript uses entropy-regularization to constraint the policy and the dynamics model. Other works in the literature already do this, however, they usually conduct their experiments with an added rewards. Here, in contrast, the reward is set to 0. The authors show that even when having no rewards interesting behavior emerges. Thus, it was interesting to see the agents’ behaviors on the proposed gird-world domains and on the cartpole.

Will the work be of significance to the field and related fields? How does it compare to the established literature? If the work is not original, please provide relevant references.

I believe the significance to the RL field might be limited in its current form, and very limited to other fields like psychology or neuroscience. Rationale below.

The mathematical framework is very similar to other proposals in the literature, as the authors already mention and cite, such as entropy-regularized and KL-regularized RL. In such frameworks one typically constrains (with the method of Lagrange multipliers) a distribution by requiring it to have either high entropy [3] or low KL distance [4] to a reference distribution. The constrained distributions typically are the policy [3,4], the transition dynamics [5] or meta-distributions over, for example, model parameters and policy in [6]. As far as I am aware the combination of regularizing both, the policy and the dynamics has only appeared in [7]. The setting in [7] is slightly more general since it's a KL regularization where the prior can be chosen at will. Given this, the methodology proposed here is strictly not very novel, however, there are 2 things that sets the current work apart. First, they consider action spaces that can change depending on the state, this makes the entropy be better suited to track states with more "action possibilities" because the KL in a sense normalizes over the action space cardinality. Second, although entropy and KL regularization are not that novel, the experiments conducted on 0 rewards on grid-worlds and cartpole are novel.

The authors motivate in the appendix that the unique measure that maximizes state-action occupancy is the entropy. It would have been nice for the authors to mention how their proof differs from the standard proof from Shannon in his original paper. And if there is any novel argument on that. I say this because parts of what I read in the appendix and the condition are very similar to Shannon's.

One of the things I didn't see mentioned in the paper is the risk-sensitivity aspect of the whole setting. The objective of the agent encourages 2 things: future cumulative policy entropies + future cumulative dynamics entropies. In other words, the agent wants to be random (in terms of actions) now and in the future, and it wants to get to states that can lead to more states in the future. It basically puts high value on stochasticity which means it is a risk-seeking agent. This is highly related to risk-sensitivity as other works have explored also with KL regularized objectives [5]. So I think the authors should explore that side of the literature and compare a bit more to it.

Similarly, related to risk-sensitivity is the noise-TV problem see reference [49] in the paper. It would have been nice that the authors compare and discuss a bit more to that problem, and extend the explanation on why the method avoids the TV problem (below I comment on the reliance on "death").

Lastly, the authors mention and cite the empowerment papers from Polani. I think this is very relevant. I would have liked a more thorough comparison and discussion with other methods in the literature that are "curiosity" driven.

Does the work support the conclusions and claims, or is additional evidence needed?

I think the experiments are ok. They show interesting behaviors obtained by maximizing this objective. However, I feel uneasy about the following 3:

1) The fact that they use entropy instead of KL is highlighted due to the benefits I mention above. However, given that the authors mention this also in the paper, it would have been nice to compare against an MDP where instead of restricting the action space the authors let the agent have all actions but if the agent moves towards a wall then it just stays in place. Then they could compare both entropy and KL method to see if there is any real difference between the two.

2) The authors seem to rely a lot on the “death” component in their experiments. That is, since the agent can “die” (enters an absorbing state and stays there) it cannot collect more entropy “rewards” in the future (either policy or dynamics entropies). Thus the agent wants to stay alive by collecting food and avoiding essentially terminal states, leading, in the process, to interesting behaviors and avoiding “noise” when it implies that the agent could die. However, what happens when the agent cannot actually die? I would assume that the agents won’t care too much about anything else other than seeking noise and stochasticity and they won’t show interesting behaviors. They will probably get stuck in a “noisy TV” since it will give a lot of future entropy.

3) The lack of comparison to other methods and the lack of experiment on larger scales other than grid-worlds and cartpole. As the authors cite, there are quite a few methods that do not use rewards. Comparing at least to a few of them, like empowerment or other information theoretic quantities (the authors cite quite a few other methods) would strengthen the paper.

Similarly, there are plenty of ways one could extend the method to neural networks. The authors cite a few of them. Showing if the method scales using standard value-estimation techniques would also convince other researchers to adopt this methodology when creating their agents.

Are there any flaws in the data analysis, interpretation and conclusions? Do these prohibit publication or require revision?

I don’t think there are any factual flaws in the analysis.

The interpretation is not entirely correct (see my comments above). I also think that the statement that the proposed objective is “the sole goal of intelligent behavior” is a bit of an overstatement.

The paper would be stronger if there were comparison to other methods that also do not use rewards.

Is the methodology sound? Does the work meet the expected standards in your field?

As mentioned before, more comparisons and experiments on bigger domains are the things one usually sees in RL papers.

Is there enough detail provided in the methods for the work to be reproduced?

I think with the information in the appendix the results could be reproduced.

[1] Silver, D., Singh, S., Precup, D., & Sutton, R.S. (2021). Reward is enough. *Artif. Intell.*, 299, 103535.

[2] Salge, Christoph, Cornelius Glackin, and Daniel Polani. "Empowerment—an introduction." *Guided Self-Organization: Inception (2014)*: 67-114.

[3] Haarnoja, Tuomas, et al. "Soft actor-critic: Off-policy maximum entropy deep reinforcement learning with a stochastic actor." *International conference on machine learning*. PMLR, 2018

[4] Fox, R., Pakman, A., and Tishby, N. Taming the noise in reinforcement learning via soft updates. In *Conference on Uncertainty in Artificial Intelligence (UAI)*, 2016

[5] Fei, Yingjie, et al. "Risk-sensitive reinforcement learning: Near-optimal risk-sample tradeoff in regret." *Advances in Neural Information Processing Systems 33 (2020)*: 22384-22395.

[6] Grau-Moya, Jordi, et al. "Planning with information-processing constraints and model uncertainty in Markov decision processes." *Machine Learning and Knowledge Discovery in Databases: European Conference, ECML PKDD 2016, Riva del Garda, Italy, September 19-23, 2016, Proceedings, Part II 16*. Springer International Publishing, 2016.

[7] Tishby, Naftali, and Daniel Polani. "Information theory of decisions and actions." *Perception-action cycle: Models, architectures, and hardware*. New York, NY: Springer New York, 2010. 601-636.

REVIEWER COMMENTS

Reviewer #1 (Remarks to the Author):

Note: We have slightly changed the terminology in the current version to make it more uniform and to avoid ambiguities. Now, we use throughout the term maximum occupancy principle (MOP), instead of principle of maximum occupation, and refer to MOP agents instead of H agents. In addition, we now exclusively refer to extrinsic and intrinsic rewards.

The authors propose state-action entropy maximization as a fundamental principle underlying the learning of behaviors. It consists in maximizing the. They exhibit on toy environments that some interesting behaviors can emerge. The structure of the paper is overall easy to follow, even though some introduction sentences could help a reader in the subsections of the method section.

Thanks a lot for the thoughtful review. We have added additional text to further motivate the different subsections of the methods section.

The significance and originality of the paper suffers from unclear conceptual and experimental differences with other methods and the lack of a clear and sound evaluation protocol, as explained below.

Thanks a lot for pointing to us these weak points, which helped us to largely improve the quality of the manuscript. Below we have addressed all your points, one by one. We have added a new comparison to other existing reward-free approaches, and we have applied MOP to large scale problems.

Major points

Unclear positioning. The paper proposes a behavior principle but lacks some discussion

about different behavior principles, like the Free-energy principle [2] or compression maximization [3].

Lack of comparison to other intrinsic motivations

Thanks for pointing out to us that the relation between the maximum occupancy principle and other theories of behavior was unclear. We have extensively rewritten the 5th paragraph of the Introduction section to better link between them and clarify the major differences. We have added the relevant references [2,3] indicated by the reviewer.

We have also added empirical comparisons between the MOP, empowered and free-energy agents in the 4-rooms and cartpole examples and found substantial differences between them, as expected (see new Sec. 3.5 and new Figure 6, along with additional Supplementary materials).

Intrinsic motivations in RL (45-53) are indeed evaluated on RL benchmarks, but they mostly aim to assess their exploration ability. The final objective is not to solve an arbitrary Atari game. In addition, the trade-off between intrinsic and extrinsic rewards is often ruled by one single hyper-parameter which could be set to 0. Therefore, it is bad faith to state that the behavioral variability ceases after learning as a general rule.

We acknowledge that this sentence was not very fortunate, and thus we have removed it.

With this in mind, numerous works can be described as maximizing state entropy (see Section 6 of [4]) and some of these methods are/can easily be associated with action-entropy maximization RL algorithms like SAC [5]. So, there already exist methods that aim to simultaneously maximize both state entropy and action entropy: what is the fundamental difference with theirs?

Indeed, many works in the RL literature have used action or state regularization mostly separately (see references in the Introduction), but none of them have explicitly combined with arbitrary weights action and state entropies. Further, to our knowledge only our current manuscript and our recent arXiv paper (now cited in the manuscript), in a different setup, does so by combining with arbitrary weights action and state entropies without any loss of mathematical tractability (see references there regarding the link with previous literature):

[2302.01098] A general Markov decision process formalism for action-state entropy-regularized reward maximization (arxiv.org)

Regarding the Sec. 6 of reference [4], it is proposed to use state entropy as a way to define novelty of visited actions. Although this looks superficially similar to our approach, we do combine action and state entropies in arbitrary ways without any loss of mathematical tractability, as said above. Further, in reference [4] the authors look for approximations to the steady-state distribution over states $p(s)$ to model the

idea of novelty, but the existence of such a steady-state distribution is not required (not even guaranteed) in our work.

At the conceptual level, there is yet another important difference between reference [4] and our work: our approach offers a new way of defining what an “interesting” state is. This does not correspond to “novel” states in the sense of low visit counts. Instead, we define “interesting” states as those that predict large future action-state path entropy, regardless of experience. This is fundamentally different from previous conceptualizations of novelty seeking and learning, as they do not seem to have formalized the idea of future visitations from those relevant states.

We have rewritten the 5th paragraph of the Introduction to better highlight the novelty of our definition of what constitutes to be an “interesting” state and how it departs from previous conceptualization of novelty seeking.

Finally, we have updated the reference [4] to the final published form.

One of the main experimental contribution of the paper is to show the emergence of particular behaviors. But it remains unclear whether a more common state entropy maximization would end up with similar outcomes.

We are happy to know that the reviewer sees in the emergence of complex behavior from MOP an interesting contribution of this manuscript. It is important to stress that other approaches have missed the important point that the presence of an internal state in the agent is important to generate variable goal-directed behaviors. For instance, in the prey-predator example, when the energy of the mouse is low, it tends to go to the food source, and when the cat is nearby, it tries to scape; otherwise, the mouse generates somehow random, lively behaviors (see extended results in Fig. 3d). These behaviors do not typically emerge from traditional approaches (as we have said, they do not use internal states). Instead, they typically generate stereotyped behaviors that are either previously designed and imposed by a handcrafted reward function (such as walking in a quadruped forced by a hand-crafted forward motion reward function in your reference [5]), or by conditioning the policy on a latent set of skills, and eventually defining tasks from an intrinsic reward (see additional clarification on the 3rd paragraph of the Discussion).

We have extended the discussion on the emergence of goal-directed behavior in reward-free settings in the 2nd paragraph of the Discussion section.

Appendix F is a first step in this direction and it should be somehow integrated in the main paper, but similar variants of the loss should also be tested in the other environments. It would also give more insights about the relative roles of the action and state entropy terms.

We agree that Supplemental Sec. F is a relevant step to show that goal-directed behavior emerges without the need of an extrinsic reward function. To complement this discussion in the main manuscript, we have now added in Fig. 2c and Fig. 3d panels showing how goal-directedness switches from moving around to food-seeking when the energy of the agent is very low.

The reviewer also suggests testing different environments, so we have implemented MOP in a quadruped, a high dimensional Gymnasium environment, and show that it generates more behavioral variability than the R agent (an epsilon-greedy survival maximizer).

Finally, we agree with the reviewer that testing the relative roles of action and state entropy terms is important. Indeed, we remind that in our previous version we have already tested several versions of the objective with different values of alpha and beta, which measure the relative contributions of action and state entropies, respectively. In Fig. 2e (right panel) we have compared the different roles of alpha and beta terms by studying how the probability of escaping the noisy room depends on beta. Interestingly, we observe that agents with large beta stay longer in the noisy room, as it provides large state entropy. In Fig. 4e, we have also studied the effect of the action and state entropies in a cartpole choosing sides to live in and found that agents with a large beta again look for regions of large state variability, depending on the noise of the controller. Finally, in Fig. 5 we exclusively study the impact of the relative weights on our agent-pet environment. Therefore, the effect of the relative strength of state and action entropy are controlled by the beta/alpha ratio, and this has been already described in three examples. We have further clarified the effect of beta in the last paragraph of Sec. 3.1 and at the end of the 4th paragraph of the Discussion, since it might have not been fully evident in the previous version of the manuscript.

Another interesting baseline could be an intrinsic reward of 1, since the positivity of the reward may induce keep-alive behaviors.

This is indeed a relevant comparison. Please, note that the R agent in two of our examples (the prey-predator and cartpole) always receives a reward of 1 whenever it is alive, and 0 otherwise, so the reviewer's suggestion has been already implemented in the previous version of the manuscript. As indicated by the reviewer, using this reward structure ($R=1$ everywhere) allows a more direct and fair comparison between MOP and R seeking behaviors, because it does not favor a priori any specific behavior (such as upright pole position in the cartpole example). As we showed in Figs. 3-4, even constant reward of 1 is not enough to generate complex enough behavior in an R agent as compared to the MOP agent: the MOP agent generates richer escaping behaviors in the prey-predator example (Fig. 3) and occupies more broadly space and generates some sort of dancing behavior in the cartpole example (Fig. 4c) than the R agent.

The result that a R agent with reward of 1 everywhere except when it dies generates a much simpler behavioral repertoire than the MOP agent can come first as a surprise. However, the rationale is clear: even in cases where action and state entropies do not depend on the state of the agent, the MOP agent has a drive to generate behaviors and move around, while the R agent with reward of 1 everywhere and 0 when death will try to find very safe regions of action-state space and stay there as long as possible without generating behavioral variability.

We have clarified this important point in the 2nd paragraph of Sec. 3.3. We have also indicated the numerical values of the rewards using digits, instead of words, as the reward structure might have not been very clearly explained in the previous version of the manuscript.

To further add on these examples in a high-dimensional, continuous control setting, we have now included our quadruped experiment with reward of 1 everywhere except when death (when it falls or starves) and compare to the MOP agent. As expected, we again find that the R agent generates way less behavioral variance than the MOP agent (see new Sec. 3.6 and new figure Fig. 7).

How are goal-directed behaviors defined? The authors claim that the agent learns goal-directed behaviors (abstract and conclusion) and it is one of the main results. I do not understand the difference between the goal-directed behaviors (e.g. food seeking) and a simple policy. In the gridworld arena of Section 3.1, we are visually biased to consider the different cells as the only state space, but since internal energy is part of the state space, what is so fantastic about the policy changing according to the energy level ?

We thank the reviewer for this important question. Although we do not have yet a formal definition of goal-directed behavior, we can tentatively argue that an agent follows a specific goal-directed behavior if, an otherwise stochastic policy, becomes deterministic when the agent lies within some region of state space. Our rationale is that a MOP agent will generate all sorts of stochastic behavior when basic energy and safety conditions are met, but when energy is low and safety is compromised, the agent will generate specific deterministic behaviors. For instance, in the 4-rooms example, the agent seems to follow some internal “goal”, such as going for food, when the internal energy is low. In the prey-predator example, the agent seems to follow a goal-directed behavior when it teases the cat to release the food area. We agree with the reviewer that this is precisely a natural, expected interpretation of our results, but it highlights the importance of defining internal states to generate interesting behaviors.

We have clarified these points with additional examples in our manuscript and by rewriting the end of the 2nd paragraph of the Discussion section. We have now added in Fig. 2 and 3 examples of behavior that deterministically emerge depending on the state of the agent (either internally or externally). Obviously, when energy is small, deterministic behavior of approaching to the food emerges (see new Fig. 2c and Fig. 3d), but also when the agent is threatened by the cat, escaping behavior emerges (Fig. 3d, right, agent avoids predator when it's not starving).

Minor points:

Limited environments: I think the paper deserves a discussion about the complexity of the environments, what happens if they progressively add more pets in 3.4 ? How could the method work with continuous states/action spaces or with larger number of states ?

The examples that have been chosen in the manuscript obey our rationale of showing tractable cases that can be solved exactly with numerical methods, without relying on function approximation of the value function or the policy, which could obscure the richness of the resulting behaviors. Our rationale was that scaling up these examples to more complex situation would be straightforward under some additional and standard approximations. This is indeed the case:

As mentioned above, we have added a large-scale problem in a Mujoco simulator consisting of a quadruped with (at least) 28 states and 8 degrees of freedom and were able to generate spontaneous behaviors such as turning, walking and jumping under MOP without the need to define a heuristic reward function that would explicitly promote them. We have also compared to an R agent with reward 1 everywhere except when it falls or starves, and find that the R agent does not generate as much variability as the MOP agent (see new Sec. 3.6, and new Fig. 7).

These results clearly show that our approach is scalable to high dimensions and continuous spaces.

Unclear equation: I do not understand how they can go from Equation 1 to Equation 2. They replaced $R(\tau)$ by equation 1, but the modified term is not the same. It looks like the expectation term should no longer be there. Please could the authors clarify that?

Note that the expectation in the last term should still be there because the entropy terms still depend on the previous a_t and s_t from where next-step action and states entropies are computed. We have now written what the expectations are over explicitly in the equations, hoping that it will clarify why the expectations still need to be there.

External vs extrinsic rewards. The authors explain they abandon external rewards in page 2, and then they explain they define an external reward function. It may come from a confusion between external rewards and extrinsic rewards. This confusion holds all along the paper and should be clarified. If they consider different definitions from the existing literature [1], it should be explicitly stated and argued for.

We thank the reviewer for noticing this important lack of consistency. We now only use the terms extrinsic and intrinsic rewards. By extrinsic rewards we mean policy-independent rewards of the form $R(s,a)$, with the function being policy-independent. This reward does not change during the course of learning. By intrinsic reward we mean policy-dependent rewards. This can change on the course of learning (e.g., improving the policy or improving a learned model of the state transitions).

We have added reference [1] in our manuscript, and we have clarified our definition of intrinsic and extrinsic rewards at the end of the 1st paragraph of Section 2.1.

Bug in prey-predator? In video 3, it happens that the cat touches the mouse, but she does not die (e.g. 6s) despite what is written in the paper. Why?

We apologize for the confusion. We define that the mouse is caught by the cat only when they exactly overlie over the same space cell at the same time. Therefore, if they cross each other, the mouse is not caught. We have clarified this in the Supplemental (see Section E.4, Transitions).

Typos:

Page 1 – introduction: Further, setting a reward function by design as the goal of artificial agents is more often than not arbitrary

Note that we have used the phrase “more often than not”, which now we write in a subclause, in order to avoid confusion.

Page 2 – introduction: action-sate

Thanks for finding it!

Page 5 – results: they confuse Figure 1 and Figure 2.

Corrected!

Others:

Please, send .mp4 video files.

We sent mp4 files but there was an unfound issue and the editorial team asked for avi files. We have submitted mp4 files again, and hopefully they will be fine.

References

[1] Oudeyer, P. Y., & Kaplan, F. (2009). What is intrinsic motivation? A typology of computational approaches. *Frontiers in neurorobotics*, 6.

[2] Friston, K., Kilner, J., & Harrison, L. (2006). A free energy principle for the brain. *Journal of physiology-Paris*, 100(1-3), 70-87.

[3] Schmidhuber, J. (2009). Driven by compression progress: A simple principle explains essential aspects of subjective beauty, novelty, surprise, interestingness, attention, curiosity, creativity, art, science, music, jokes. In *Anticipatory Behavior in Adaptive Learning Systems: From Psychological Theories to Artificial Cognitive Systems 4* (pp. 48-76). Springer Berlin Heidelberg.

[4] Aubret, A., Matignon, L., & Hassas, S. (2023). An information-theoretic perspective on intrinsic motivation in reinforcement learning: a survey. *Entropy*, 25(2), 327.

[5] Haarnoja, T., Zhou, A., Abbeel, P., & Levine, S. (2018, July). Soft actor-critic: Off-policy maximum entropy deep reinforcement learning with a stochastic actor. In *International conference on machine learning* (pp. 1861-1870). PMLR.

Reviewer #2 (Remarks to the Author):

Note: We have slightly changed the notation in the current version to make our terminology more uniform and to avoid ambiguities. Now, we use throughout the term maximum occupancy principle (MOP), instead of principle of maximum occupation, and refer to MOP agents instead of H agents.

The paper presents an MDP framework with known transition dynamics where instead of considering the maximization of an external reward signal it is considered the maximization of action entropy and future state entropy. This objective is proposed as “the sole goal of intelligent behavior” and as a proxy for obtaining curiosity-driven behavior. Theoretical justification is provided for this objective, and experiments on small domains are conducted showing the applicability of the methodology.

What are the noteworthy results?

There is currently a hot debate on what are rewards in RL and where do they come from [1]. The proposal of not using rewards to obtain seemingly intelligent policies, as proposed by this work and others[2] is interesting in its own right.

We thank the reviewer for the positive assessment and thoughtful review of our manuscript, and thanks for the comment, which have served to improve our manuscript.

The manuscript is well written and easy to understand. There has been a significant effort to clearly explain the experiments and the mathematical statements.

The present manuscript uses entropy-regularization to constraint the policy and the dynamics model. Other works in the literature already do this, however, they usually conduct their experiments with an added rewards. Here, in contrast, the reward is set to 0. The authors show that even when having no rewards interesting behavior emerges. Thus, it was interesting to see the agents’ behaviors on the proposed gird-world domains and on the cartpole.

Will the work be of significance to the field and related fields? How does it compare to the established literature? If the work is not original, please provide relevant references.

I believe the significance to the RL field might be limited in its current form, and very limited to other fields like psychology or neuroscience. Rationale below.

Below we address all the comments in detail.

The mathematical framework is very similar to other proposals in the literature, as the

authors already mention and cite, such as entropy-regularized and KL-regularized RL. In such frameworks one typically constrains (with the method of Lagrange multipliers) a distribution by requiring it to have either high entropy [3] or low KL distance [4] to a reference distribution. The constrained distributions typically are the policy [3,4], the transition dynamics [5] or meta-distributions over, for example, model parameters and policy in [6]. As far as I am aware the combination of regularizing both, the policy and the dynamics has only appeared in [7]. The setting in [7] is slightly more general since it's a KL regularization where the prior can be chosen at will.

Thanks for pointing to us the relevant reference [7], which is now cited, like the other missing papers.

First, our understanding of [7] slightly differs from that of the reviewer, if we may: there is a KL regularization in Eq. 21 in [7], but the priors $p(s_t)$ and $\pi(a_t)$ need to be consistent with the conditional probability distributions $p(s_{t+1}|s_t, a_t)$ and $\pi(a_t|s_t)$, and therefore they cannot be arbitrarily chosen. Further, as the priors need to be probability distributions, when they appear in the denominator the number of actions and states are regularized as well, and therefore it will incur into the same problem commented in our manuscript that this objective does not favor states with large number of actions and future states. Therefore, the objective in Eq. 21 is substantially different from (weighted) action-state path maximization if the number of actions depends on the state. We agree, however, with the reviewer that one can formally choose in Eq. 21 any number in the denominators and the equations would still be valid, but this is not what the authors propose to optimize.

Second, our framework can also include default policies with no change in the proofs and it changes the equations as mentioned in the last paragraph of Section 2.2.

Finally, ref [7] does not contemplate any arbitrary mixture of action and state entropies (weighted by the parameters α and β in our manuscript), and therefore the framework in [7] is not as general as ours. We would like to mention that a close approach to that in our current manuscript is our following arXiv paper

[\[2302.01098\] A general Markov decision process formalism for action-state entropy-regularized reward maximization \(arxiv.org\)](https://arxiv.org/abs/2302.01098)

whose Introduction may be relevant to see that previous approaches have not considered arbitrary mixtures of action and state entropies.

We have commented on all these important questions in the new 4th paragraph of the Discussion Section.

Given this, the methodology proposed here is strictly not very novel, however, there are 2 things that sets the current work apart. First, they consider action spaces that can change depending on the state, this makes the entropy be better suited to track states with more "action possibilities" because the KL in a sense normalizes over the action space cardinality. Second, although entropy and KL regularization are not that novel, the experiments conducted on 0 rewards on grid-worlds and cartpole are novel

We agree this is a very accurate description of our contribution. We have further stressed the new aspects of our methodology in the new version of the manuscript, as they are some novel contributions that are not present in the previous literature: (1) an arbitrary combination of action and state entropies (see previous comment and the 4th paragraph of the Discussion), and (2) a proof that path entropy is the only reasonable measure of path occupancy (see next comment).

The authors motivate in the appendix that the unique measure that maximizes state-action occupancy is the entropy. It would have been nice for the authors to mention how their proof differs from the standard proof from Shannon in his original paper. And if there is any novel argument on that. I say this because parts of what I read in the appendix and the condition are very similar to Shannon's.

We thank the reviewer for this suggestion. Indeed, the proof we show in the Supplemental is quite similar to Shannon's information measure. However, there are subtle differences, which we have highlighted now in a Remark in Section A of the Supplemental.

One of the things I didn't see mentioned in the paper is the risk-sensitivity aspect of the whole setting. The objective of the agent encourages 2 things: future cumulative policy entropies + future cumulative dynamics entropies. In other words, the agent wants to be random (in terms of actions) now and in the future, and it wants to get to states that can lead to more states in the future. It basically puts high value on stochasticity which means it is a risk-seeking agent. This is highly related to risk-sensitivity as other works have explored also with KL regularized objectives [5]. So I think the authors should explore that side of the literature and compare a bit more to it.

Thanks a lot for this very good point. We have added at the end of the 2nd paragraph in the Discussion a comment about risk-seeking behavior and how our framework can be interpreted to favor it. Nevertheless, note that the fact that we have terminal states, and that agents avoids them, also acts against risk-seeking behavior in the strict sense.

[Note: We took the paragraph from the reviewer that was initially here and inserted it further below, given the related discussion]

Lastly, the authors mention and cite the empowerment papers from Polani. I think this is very relevant. I would have liked a more thorough comparison and discussion with other methods in the literature that are "curiosity" driven.

Thanks a lot for this important comment. We have rewritten the 5th paragraph in the Introduction to better discuss the differences between the groundbreaking work of Polani et al and MOP.

Further, we have added simulations of the 4-room gridworld and cartpole experiments to compare with empowerment approaches (new Section 3.5 and new Figure 6). As expected, and as previously reported (Jung et al. (2011)), we find that

the empowered cartpole gets stuck in the upright position and stops generating variability.

Does the work support the conclusions and claims, or is additional evidence needed?

I think the experiments are ok. They show interesting behaviors obtained by maximizing this objective. However, I feel uneasy about the following 3:

1) The fact that they use entropy instead of KL is highlighted due to the benefits I mention above. However, given that the authors mention this also in the paper, it would have been nice to compare against an MDP where instead of restricting the action space the authors let the agent have all actions but if the agent moves towards a wall then it just stays in place. Then they could compare both entropy and KL method to see if there is any real difference between the two.

Thanks again for the question. In the new Supplemental Sec. F and Fig. F.5, we have compared, using the 4-room gridworld, a MOP agent having fixed number of actions for non-terminal states (where state transitions are limited by the walls, such that moving into wall doesn't change the location), our standard R agent with fixed number of actions, and a KL divergence minimizer with a default policy that is uniform over actions. As we show graphically and mathematically, having a default policy regularizes the number of actions, which makes the KL divergence a random walker over all types of states. This means that a KL agent dies much quicker and thus it does not capture the behaviors we have shown in the manuscript, due to its insensitivity to the number of actions. As the reviewer points out, having a fixed number of actions over non-terminal states does affect the behavior of the MOP agent, which still avoids terminal states, but now does not prefer the center of the rooms and in fact more uniformly covers the arena. In conclusion, using a KL objective without extrinsic rewards leads to a uniform policy, regardless of the presence of terminal states, leading to uninteresting behaviors.

2) The authors seem to rely a lot on the "death" component in their experiments. That is, since the agent can "die" (enters an absorbing state and stays there) it cannot collect more entropy "rewards" in the future (either policy or dynamics entropies). Thus the agent wants to stay alive by collecting food and avoiding essentially terminal states, leading, in the process, to interesting behaviors and avoiding "noise" when it implies that the agent could die. However, what happens when the agent cannot actually die? I would assume that the agents won't care too much about anything else other than seeking noise and stochasticity and they won't show interesting behaviors. They will probably get stuck in a "noisy TV" since it will give a lot of future entropy.

Related: Similarly, related to risk-sensitivity is the noise-TV problem see reference [49] in the paper. It would have been nice that the authors compare and discuss a bit more to that problem, and extend the explanation on why the method avoids the TV problem (below I comment on the reliance on "death").

This is again a good comment. Indeed, if the agent cannot die, then it will generate a random-walk like behavior (as we mentioned in the first version of the manuscript at the beginning of Sec. 3.1), and we agree with the reviewer that in this particular case the agent will stay in the noisy TV room forever. This is of course a very uninteresting behavior, but it will be the optimal behavior for an immortal agent. This highlights the importance of both terminal states but also internal states: both types of states fundamentally shape the behavior of the agent, which is how we think we can model naturalistic behavior –behavior is subject to tight mechanical and energetic constraints. In other words, the presence of the constraints given by the terminal and internal states is critical for the generation of variable goal-directed behaviors, and without them there is little possibility of generating any interesting, goal-directed behavior.

To clarify this important point, we have rewritten the 2nd paragraph of the Discussion.

3) The lack of comparison to other methods and the lack of experiment on larger scales other than grid-worlds and cartpole. As the authors cite, there are quite a few methods that do not use rewards. Comparing at least to a few of them, like empowerment or other information theoretic quantities (the authors cite quite a few other methods) would strengthen the paper.

Similarly, there are plenty of ways one could extend the method to neural networks. The authors cite a few of them. Showing if the method scales using standard value-estimation techniques would also convince other researchers to adopt this methodology when creating their agents.

The paper would be stronger if there were comparison to other methods that also do not use rewards.

Is the methodology sound? Does the work meet the expected standards in your field?

As mentioned before, more comparisons and experiments on bigger domains are the things one usually sees in RL papers.

Thanks a lot for these comments. The examples that have been chosen in the manuscript obey our rationale of showing tractable cases that can be solved exactly with numerical methods, without relying on function approximation of the value function or the policy, which could obscure the richness of the resulting behaviors. Our rationale was that scaling up these examples to more complex situation would be straightforward under some additional and standard approximations. This is indeed the case: We have added a large-scale problem in a Mujoco simulator consisting of a quadruped with (at least) 28 states and 8 degrees of freedom and were able to generate spontaneous behaviors such as turning, walking and jumping under MOP without the need to define a heuristic reward function that would explicitly promote them (see new Sec. 3.6 and new Figure 7). We have also compared to an R agent with reward 1 everywhere except when it falls or starves, and find that the R agent does not generate as much variability as the MOP agent. These results

clearly show that our approach is scalable to high dimensions and continuous spaces.

Following the reviewer's second recommendation, and as mentioned above, we have also added simulations of the 4-room gridworld and cartpole experiments to compare with empowerment and free energy approaches (see new Sec. 3.5 and new Fig. 6, along with several Supplementary materials). As expected, and as previously reported by other authors (see references), we find that the empowered cartpole gets stuck in the upright position and stops generating much variability, and free energy is equivalent to a reward maximizer that does not show any variability.

Are there any flaws in the data analysis, interpretation and conclusions? Do these prohibit publication or require revision?

I don't think there are any factual flaws in the analysis. The interpretation is not entirely correct (see my comments above). I also think that the statement that the proposed objective is "the sole goal of intelligent behavior" is a bit of an overstatement.

Thanks for this comment. We agree that this sentence is very not fortunate, so we have rewritten that sentence and we have also removed all instances of "sole".

Is there enough detail provided in the methods for the work to be reproduced?

I think with the information in the appendix the results could be reproduced.

[1] Silver, D., Singh, S., Precup, D., & Sutton, R.S. (2021). Reward is enough. *Artif. Intell.*, 299, 103535.

[2] Salge, Christoph, Cornelius Glackin, and Daniel Polani. "Empowerment—an introduction." *Guided Self-Organization: Inception (2014)*: 67-114.

[3] Haarnoja, Tuomas, et al. "Soft actor-critic: Off-policy maximum entropy deep reinforcement learning with a stochastic actor." *International conference on machine learning*. PMLR, 2018

[4] Fox, R., Pakman, A., and Tishby, N. Taming the noise in reinforcement learning via soft updates. In *Conference on Uncertainty in Artificial Intelligence (UAI)*, 2016

[5] Fei, Yingjie, et al. "Risk-sensitive reinforcement learning: Near-optimal risk-sample tradeoff in regret." *Advances in Neural Information Processing Systems 33 (2020)*: 22384-22395.

[6] Grau-Moya, Jordi, et al. "Planning with information-processing constraints and model uncertainty in Markov decision processes." *Machine Learning and Knowledge Discovery in*

Databases: European Conference, ECML PKDD 2016, Riva del Garda, Italy, September 19-23, 2016, Proceedings, Part II 16. Springer International Publishing, 2016.

[7] Tishby, Naftali, and Daniel Polani. "Information theory of decisions and actions." Perception-action cycle: Models, architectures, and hardware. New York, NY: Springer New York, 2010. 601-636.

REVIEWER COMMENTS

Reviewer #1 (Remarks to the Author):

I thank the authors for their work. The rewriting, responses, and new comparisons clarified some of my concerns and misunderstandings. I will comment on some parts of their answer.

"None of them have explicitly combined with arbitrary weights action and state entropies."

Again, that is just a matter of hyper-parameters that can most of the time be adjusted. Every paper working on SAC hyper-parameterizes the action entropy parameter, even though the objective is not the same as for this paper.

"It is important to stress that other approaches have missed the important point that the presence of an internal state in the agent is important to generate variable goal-directed behaviors. [...] they typically generate stereotyped behaviors that are either previously designed and imposed by a hand-crafted reward function"

In previous approaches on intrinsic motivation ([62] in your current paper), the presence of an internal state in the RL environment is orthogonal to their proposal and they are compatible with 0-reward environments. I got your points and your differences with respect to them, but as they also aim to generate variability, I think comparing the resulting behaviors make sense. In addition, their intrinsic rewards is often strictly positive, which should induce keep-alive behaviors. For your tabular environments, I think [1] could be used.

- The reviewer also suggests testing different environments, so we have implemented MOP in a quadruped, a high dimensional Gymnasium environment, and show that it generates more behavioral variability than the R agent (an epsilon-greedy survival maximizer).

You implemented a widespread existing baseline without any changes and found well-known results: action entropy induces variability (I think we all agree it is not new) and positive (intrinsic) rewards induce keep-alive behaviors ([1] for an example of the latter point). To do that, you set beta equal to zero, which is supposed to be one of the most important thing of your contribution. I don't think that your answer "These results clearly show that our approach is scalable to high dimensions and continuous spaces." holds here. A better fit for your contribution could be a stochastic environment where you use $\beta > 0$, but it seems more difficult to do.

Overall:

There are two parts in the proposed reward function, action entropy and next-state entropy.

- The authors show that most of the variability of the behavior as well as its "goal-conditioned" keep-alive part come from the action entropy, which is a well-known result: 1) the gridworld and ant environments do not leverage the next-state entropy. 2) For the prey-predator and pendulum, there is no analysis of the impact of beta on the original environment. 3) The pet-fence is the only "original" environment where next-state entropy seems to matter. Nonetheless, it remains unclear whether other state-of-the-art intrinsic motivations generate the same or more variability while keeping the agent alive.

- The authors study the impact of next-state entropy with respect to noisy state transitions in the environment. However, since one of the objective is "we have proposed that a major goal of intelligence is to 'occupy path space'", it lacks strong arguments about whether intelligent systems are indeed inherently attracted by noise. I'm a bit skeptical about the potential of such argument, but it may be an original direction of work as most of intrinsic motivations currently aim to avoid such noise.

- There are several contributions here and there: creating an environment with an internal state + 0 reward and observe what the authors call "goal-directed" behaviors or the mathematical properties of the action-state path occupancy. I think addressing one of two above-mentioned points would crucially strengthen the significance of the contribution. Addressing these points need significant work on the manuscript.

The claims, the analysis, interpretations are ok. I explained my concerns about the methodology, significance and novelty of the work. One should be able to reproduce the work.

[1] Strehl, A. L., & Littman, M. L. (2008). An analysis of model-based interval estimation for Markov decision processes. *Journal of Computer and System Sciences*, 74(8), 1309-1331.

[2] Eysenbach, B., Gupta, A., Ibarz, J., & Levine, S. (2018, September). Diversity is All You Need: Learning Skills without a Reward Function. In *International Conference on Learning Representations*.

Reviewer #2 (Remarks to the Author):

I want to thank the authors for the considerable amount of work conducted between revisions.

The paper has clearly improved. On one hand the discussion and positioning with respect to similar works has been enhanced and I believe it is fairer. In addition, the authors have stated the importance of absorbing states and internal states for the methodology to really show interesting behaviors. I still find this not fully satisfactory enough since, in my opinion, any intrinsic motivation scheme should still work (i.e. show interesting behaviors) in the absence of absorbing states (i.e. I would expect an intelligent agent that is immortal, to still show interesting behavior and not get stuck on noisy TVs). However, with the assumption on death and absorbing states, the methodology seems compelling to me. On the other hand, the experimental setting has clearly improved by comparing to other methods and baselines (e.g. empowerment) and by testing on slightly more complex motor-control problems like the ant environment using neural networks.

In conclusion, given the above improvements and changes to the discussion, I recommend for acceptance.

Please see below our responses (indented and blue) to reviewers' comments (cursive).

REVIEWER COMMENTS

Reviewer #1 (Remarks to the Author):

I thank the authors for their work. The rewriting, responses, and new comparisons clarified some of my concerns and misunderstandings. I will comment on some parts of their answer.

"None of them have explicitly combined with arbitrary weights action and state entropies." Again, that is just a matter of hyper-parameters that can most of the time be adjusted. Every paper working on SAC hyper-parameterizes the action entropy parameter, even though the objective is not the same as for this paper.

Thanks for this comment. What we meant is that our work is novel regarding the mixing of action and state entropies in any arbitrary way. We have shown the effect of the alpha and beta parameters in several examples of the paper (gridworld, cartpole and pet). We agree with the reviewer that SAC has alpha as a parameter. However, 1) it does not have beta, the weight for next-state entropy, and 2) in SAC, alpha is typically interpreted as a 'temperature' parameter, given that it has been used only so far in high-dimensional spaces to maximize extrinsic reward. Instead, alpha in MOP should now be taken relative to beta, which makes alpha acquire a novel meaning, different than 'temperature'. Let us add that using SAC or any other algorithm in our high-dimensional problems is not core to our paper; what is core is the notion of reward-free entropy seeking agents.

"It is important to stress that other approaches have missed the important point that the presence of an internal state in the agent is important to generate variable goal-directed behaviors. [...] they typically generate stereotyped behaviors that are either previously designed and imposed by a hand-crafted reward function."

In previous approaches on intrinsic motivation ([62] in your current paper), the presence of an internal state in the RL environment is orthogonal to their proposal and they are compatible with 0-reward environments.

If we understand correctly, the reviewer agrees with us in that in previous approaches of intrinsic motivation did not consider internal states as fundamental to generate goal-directed behavior. Using internal states lets agents model the external world with respect to their needs, and paired with the maximum occupancy principle, it generates complex behavior. We have further clarified this at the end of the second paragraph in the Discussion.

I got your points and your differences with respect to them, but as they also aim to generate variability, I think comparing the resulting behaviors make sense. In addition, their intrinsic rewards is often strictly positive, which should induce keep-alive behaviors.

Ref. [62] in our paper is a review that categorizes many intrinsic motivation approaches in RL, and we have already addressed the differences with the relevant ones in the Introduction and Discussion. Furthermore, we have already directly compared MOP with the two most standard reward-free approaches, MPOW and FEP, so we feel that going into a further exploration and comparison would only obscure our results. We have shown in our paper that these standard approaches are not competitive with respect MOP in generating variability and different behaviors.

Note that we have already tried the case of $R=1$ when alive, and $R=0$ when dead (see R agents for the predator-prey, cartpole and ant experiments), and we have already seen that this strictly positive intrinsic reward does generate keep-alive behaviors, at the expense of producing little behavioral variability. When adding a standard stochastic selection of actions (epsilon-greedy), MOP still generates higher variability even when matching average lifetimes.

For your tabular environments, I think [1] could be used.

Thanks for bringing this paper to our attention. In ref [1] an efficient method using interval estimations is developed to resolve the exploration-exploitation tradeoff in a PAC-like manner. This means that they focus on the learning problem, where reward and state transition distributions are not known. This motivates the action selection problem to either explore a new action, or exploit a known action, based on the uncertainty about their consequences (interval estimation). The variability of these algorithms should decrease as learning proceeds, given that both the interval estimation and the exploration bonus converge in static MDPs after extensive interaction, which is very different from our agents that keep generating variability even if the world is known from the start. If we chose this algorithm and used a positive reward, like 1 for being alive (and 0 otherwise), then the algorithm from [1] will be PAC guaranteed to, at some point, behave like the R agents defined in the manuscript, who have access to the world model and behave optimally with respect to this objective. Adding learning would be an interesting step that we plan to tackle in the future, but has not been the focus of our current work.

- The reviewer also suggests testing different environments, so we have implemented MOP in a quadruped, a high dimensional Gymnasium environment, and show that it generates more behavioral variability than the R agent (an epsilon-greedy survival maximizer).

You implemented a widespread existing baseline without any changes and found well-known results: action entropy induces variability (I think we all agree it is not new) and

positive (intrinsic) rewards induce keep-alive behaviors ([1] for an example of the latter point).

We are sorry that our narrative was not clear enough. First, we would like to point out that the ant baseline that we have used has been adapted by us to include internal energy in Fig. 7e-g, so our environment is indeed novel. Further, with this example, we have shown not only that the MOP agent can generate variable behavior, but, more importantly, that it can switch from running to moving around the food source. Finally, remember that we have shown that positive reward $R=1$ for being alive and $R=0$ for dead does indeed favor living behavior, but not behavioral variability at all (see our Fig. 7 and Supplemental Fig. E.4)

To do that, you set beta equal to zero, which is supposed to be one of the most important thing of your contribution. I don't think that your answer "These results clearly show that our approach is scalable to high dimensions and continuous spaces." holds here. A better fit for your contribution could be a stochastic environment where you use $\beta > 0$, but it seems more difficult to do.

Thanks a lot for this comment. To show that our contribution is indeed (fully) scalable, we followed the reviewer's suggestion and have implemented $\beta > 0$ for the challenging ant experiment (see new paragraph in Supplemental Sec. E.7, and added paragraph at the end of Sec. 3.6). As expected, when adding stochasticity to the ant environment in the form of state transition noise only in a half of the arena ($x > 0$), the MOP ant with $\beta > 0$ tends to prefer that half, whereas the MOP ant $\beta = 0$ does not show a preference (Supplemental Fig. E.5). Importantly, if the noise is too high, the MOP ant will not go to the stochastic half of the arena due to the risk of dying, and this preference is modulated by β as shown in the figure.

Overall:

There are two parts in the proposed reward function, action entropy and next-state entropy. - The authors show that most of the variability of the behavior as well as its "goal-conditioned" keep-alive part come from the action entropy, which is a well-known result

We hope that we have convinced the reviewer with several examples, including the new one described above (ant with $\beta > 0$), and the comparison with other established reward-free approaches, that our framework is unique at generating behavior in complex agents that are (1) variable, and (2) that are goal-directed, and (3) behaviors change depending on internal states. To our knowledge, no other previous framework has shown that maximizing path entropy leads to interesting behaviors without providing any extrinsic reward signal or specific task instructions, nor how α and β differently modulate behavior.

: 1) the gridworld and ant environments do not leverage the next-state entropy. 2) For the prey-predator and pendulum, there is no analysis of the impact of beta on the original environment. 3) The pet-fence is the only "original" environment where next-state entropy seems to matter.

We are sorry that there must be some basic confusion here, because in Fig. 2e we study the effect of beta in the gridworld, and in Fig. 4e-f we study the effect of beta in the cartpole. Let us know whether this was not clear, and we will do our best to clarify this in the manuscript.

Nonetheless, it remains unclear whether other state-of-the-art intrinsic motivations generate the same or more variability while keeping the agent alive.

Again, it seems to be some basic confusion here, as in Fig. 6 we have now compared MOP with the two most standard reward-free, intrinsic motivations approaches: Empowerment and the Free Energy Principle. Again, let us know whether somehow this was not clear.

- The authors study the impact of next-state entropy with respect to noisy state transitions in the environment. However, since one of the objective is "we have proposed that a major goal of intelligence is to 'occupy path space'", it lacks strong arguments about whether intelligent systems are indeed inherently attracted by noise. I'm a bit skeptical about the potential of such argument, but it may be an original direction of work as most of intrinsic motivations currently aim to avoid such noise.

Thanks a lot for the comment. Under certain conditions, it has been shown that humans are indeed attracted by noise (Ref. [97] in the previous submission), but we agree with the reviewer that more work needs to be done empirically to study the circumstances in which agents do get attracted by noise. One of the key contributions of our work, however, is that this attraction can be modulated by beta, and that MOP agents can be either attracted or repulsed by noise, depending on beta and the effect of state transition noise on their expected lifetimes (see Fig. 2e and Supplemental Fig. E.5).

- There are several contributions here and there: creating an environment with an internal state + 0 reward and observe what the authors call "goal-directed" behaviors or the mathematical properties of the action-state path occupancy.

Thanks for the comment. We agree with the reviewer that these are two contributions from our work that fill a gap in the literature.

I think addressing one of two above-mentioned points would crucially strengthen the significance of the contribution. Addressing these points need significant work on the manuscript.

Thanks for the suggestion. As described above, we have followed it by addressing the effect of state transition noise (through beta) in the challenging ant environment.

The claims, the analysis, interpretations are ok. I explained my concerns about the methodology, significance and novelty of the work. One should be able to reproduce the work.

[1] Strehl, A. L., & Littman, M. L. (2008). An analysis of model-based interval estimation for Markov decision processes. *Journal of Computer and System Sciences*, 74(8), 1309-1331.

[2] Eysenbach, B., Gupta, A., Ibarz, J., & Levine, S. (2018, September). Diversity is All You Need: Learning Skills without a Reward Function. In *International Conference on Learning Representations*.

Reviewer #2 (Remarks to the Author):

I want to thank the authors for the considerable amount of work conducted between revisions.

The paper has clearly improved. On one hand the discussion and positioning with respect to similar works has been enhanced and I believe it is fairer. In addition, the authors have stated the importance of absorbing states and internal states for the methodology to really show interesting behaviors. I still find this not fully satisfactory enough since, in my opinion, any intrinsic motivation scheme should still work (i.e. show interesting behaviors) in the absence of absorbing states (i.e. I would expect an intelligent agent that is immortal, to still show interesting behavior and not get stuck on noisy TVs).

We see the point of the reviewer, and it is well taken. We will elaborate more on this in future research.

However, with the assumption on death and absorbing states, the methodology seems compelling to me. On the other hand, the experimental setting has clearly improved by comparing to other methods and baselines (e.g. empowerment) and by testing on slightly more complex motor-control problems like the ant environment using neural networks.

In conclusion, given the above improvements and changes to the discussion, I recommend for acceptance.

REVIEWER COMMENTS

Reviewer #1 (Remarks to the Author):

I thank the authors for their answers and modifications.

"we have now compared MOP with the two most standard reward-free, intrinsic motivations approaches: Empowerment and the Free Energy Principle."

MPOW and FEP do not aim to generate variability. These are computational principles with arguably other biologically relevant properties.

"If we chose this algorithm and used a positive reward, like 1 for being alive (and 0 otherwise), then the algorithm from [1] will be PAC guaranteed to, at some point, behave like the R agents defined in the manuscript". Indeed, in this case, the variability from time 0 to T may matter. The comparison may be irrelevant mostly if the authors manage to argue that variability likely (or should?) remains after asymptotic learning in biological systems. Otherwise, it is unclear why a specific kind of variability would be more relevant than another one for modelling "the goal of behavior".

"We are sorry that there must be some basic confusion here, because in Fig. 2e we study the effect of beta in the gridworld, and in Fig. 4e-f we study the effect of beta in the cartpole."

As far as I understand, the authors modified the original cartpole environment to introduce stochasticity. That is why I specified "original". Both the stochastic gridworld and the ant environments are also hand-crafted to showcase properties of the algorithm (and not properties of the real world). It makes it difficult to catch the general interest of beta, beyond the considered environments.

I still have concerns about the significance of this work. Except for a single citation late in the paper, there are no arguments that support the importance of targeting inherently stochastic areas. The authors could as well drop the beta-part from the loss function, remove the evaluation on hand-crafted stochastic environment and leave the paper as it is. It would change nothing with respect to the objective of the paper. But in this case, it results in an already-implemented principle which is not compared to important previous methods (I'm not convinced by the irrelevance of the comparison). When I say arguments, it could simply be intuitive examples and a discussion of behaviors in biological systems.

Overall, what does this principle may explain about intelligent behaviors of biological systems that others don't ? I don't find a clear answer to that question. Most comparison baselines show keep-alive behaviors, raw variability is not fairly compared and targeting stochastic areas does not sound important.

Since the authors addressed none of my two main concerns, raised in the previous iteration (rigorous variability comparison or strong arguments for the importance of targeting stochastic areas), I recommend a reject.

Reviewer #2 (Remarks to the Author):

I am gonna add my point of view to the main concerns raised from Reviewer 1:

> I still have concerns about the significance of this work. Except for a single citation late in the paper, there are no arguments that support the importance of targeting inherently stochastic areas.

The authors could as well drop the beta-part from the loss function, remove the evaluation on hand-crafted stochastic environment and leave the paper as it is. It would change nothing with respect to the objective of the paper.

Thank you for raising awareness about this. I want to give my arguments on why we shouldn't rush into dismissing the paper because of this. As you mention later, maybe simply laying out the arguments is enough, and we should give the opportunity for the authors to do so. Getting to the bottom of this concern would be a whole series of papers in itself as far as I can see (it is a tough one I believe).

Statement 1: Nature is stochastic from the point of view of agents with limited resources and information.

Reasons to believe this is true:

* Even if nature is assumed to be deterministic, humans don't have the knowledge about all the latent variables involved in the mechanisms that affect the dynamics of the world. This makes humans have partial observability about the world and many things appear stochastic to us (due to marginalization of beliefs).

* Due to chaos, nature is unpredictable even if deterministic when having some uncertainty in the initial conditions (i.e. always), even if this uncertainty is infinitesimal.

Implications of Statement 1: The authors modifying the ant and cartpole environment to exhibit stochasticity is actually making them more similar to what we would expect from nature. So, the “original” environments are not “better” representatives as claimed by Reviewer 1.

Statement 2: Intelligent beings (humans and animals) exhibit risk-sensitivity, i.e. sensitivity to uncertainty.

Reasons to believe this and further thoughts:

* There is a huge body of literature on risk-sensitivity in humans and in RL agents (e.g. see Braun et al below). And it is extremely clear and accepted in the field that humans are risk-sensitive. Even more, they can be risk-averse in some situations and risk-seeking in others.

* Why are humans risk-sensitive? That is a tougher question to ask and as far as I understand there is no clear agreement in the community. The most reasonable one for me seems to be about boosting robustness (risk-averse) or boosting exploration (risk-seeking) when needed. Still, it is unclear how humans decide to boost one or the other. Another (likely) possibility is that these kinds of “biases” appear due to contextual situations reinforced by the experiences of the agent.

* Even in the case of nature being determinism, one can show that risk sensitivity can arise in 2 player games (see Grau-Moya 2022 below) purely from data if players are allowed to modify their policies over time.

Implications of Statement 2: Going back to your comment, dropping the beta completely would simply hinder the model capabilities to model risk-seeking behavior (that boosts exploration of states). What the authors do in their experiments is to show that they can model risk-sensitivity when there is noise in fixed environments (a requisite for risk-sensitivity, since deterministic environments can't exhibit risk if they are static), that's why they needed to modify the “original” environments. Since risk-sensitivity is ubiquitous in intelligent beings as mentioned above, being able to model it is a plus in my view. Since beta modulates the risk-sensitivity strength, an agent that can control beta can control whether to get stuck in a noise TV or not (though this case is not considered in the paper, I can imagine one could implement such an agent).

> But in this case, it results in an already-implemented principle which is not compared to important previous methods (I'm not convinced by the irrelevance of the comparison). When I say arguments, it could simply be intuitive examples and a discussion of behaviors in biological systems.

My 2 cents on this:

* In my view the authors did provide a comparison to other non-reward methods as requested (empowerment and the free energy principle). If Reviewer 1 knows about other methods that would be a better fit or more "relevant" to the presented methodology, it would be great to mention it in the review so that the authors can do something about it and provide an analysis.

* I completely agree that laying out the arguments and intuitive examples would be something the authors can do.

> Overall, what does this principle may explain about intelligent behaviors of biological systems that others don't ? I don't find a clear answer to that question. Most comparison baselines show keep-alive behaviors, raw variability is not fairly compared and targeting stochastic areas does not sound important.

* In my view it provides a different explanation to the emergence of interesting behavior using a simple reward-free objective, which happens to also capture risk-sensitive behavior. The comparison to other reward-free methods is there (empowerment and free energy). Maybe it is true that the message can be rewritten in a clearer way.

* RE variability. I am not sure why we need this comparison?

* RE targeting stochastic areas: see Statement 2.

Conclusions:

* I would let Reviewer 1 mention the other relevant methods so that authors can add the comparison.

* I would encourage the authors to clarify the story and explain the concerns of Reviewer 1 wrt stochasticity.

References:

Risk-sensitivity in sensorimotor control

Daniel A. Braun^{1,2*} Arne J. Nagengast^{1,3} Daniel M. Wolpert¹

<https://www.frontiersin.org/articles/10.3389/fnhum.2011.00001/full>

Grau-Moya, Jordi, et al. "Beyond Bayes-optimality: meta-learning what you know you don't know." arXiv preprint arXiv:2209.15618 (2022).

Here you will find our response (indented and in blue) to the reviewer's comments (italics, unindented).

REVIEWER COMMENTS

Reviewer #1 (Remarks to the Author):

I thank the authors for their answers and modifications.

We highly appreciate the willingness of the reviewer to improve the quality of the manuscript by pointing to the important missing discussion about the presence of variability after learning in living systems and the role of risk-sensitivity in behavior.

We have answered all your comments below and added two new paragraphs in the Discussion, as well as additional context in the second paragraph in the Introduction (please see document of tracked changes for more details). We hope that these additions would make clearer the relevance of MOP.

"we have now compared MOP with the two most standard reward-free, intrinsic motivations approaches: Empowerment and the Free Energy Principle."

MPOW and FEP do not aim to generate variability. These are computational principles with arguably other biologically relevant properties.

We agree that neither MPOW nor FEP allow for stochastic policies. However, they are both established reward-free principles of behavior that have either conceptual or mathematical overlap with MOP, which is why a direct comparison was warranted and fairly asked for in a previous round of review. A priori, some ingredients of MPOW and FEP could be thought of explaining or including MOP, and our comparison aims at highlighting the differences between these frameworks, one of which is the lack of variability in both FEP and MPOW, a hallmark of living systems.

"If we chose this algorithm and used a positive reward, like 1 for being alive (and 0 otherwise), then the algorithm from [1] will be PAC guaranteed to, at some point, behave like the R agents defined in the manuscript".

Indeed, in this case, the variability from time 0 to T may matter.

We agree with this statement that the speed of learning (convergence speed to the optimal policy) depends on the type of noise and exploration strategies that agents undergo before full learning. This is non-controversial, as it is well-known that after learning in any pure reward-maximizing problem there is a deterministic optimal policy that best solves the problem. Therefore, we proceed following your recommendation below to address why asymptotic variability is expected to be present in natural agents even after learning.

The comparison may be irrelevant mostly if the authors manage to argue that variability likely (or should?) remains after asymptotic learning in biological

systems. Otherwise, it is unclear why a specific kind of variability would be more relevant than another one for modelling "the goal of behavior".

Thanks for the suggestion. We have extended our Discussion to argue that asymptotic variability is expected in living systems even in conditions where tasks are known and/or learnt (3rd paragraph).

"We are sorry that there must be some basic confusion here, because in Fig. 2e we study the effect of beta in the gridworld, and in Fig. 4e-f we study the effect of beta in the cartpole."

As far as I understand, the authors modified the original cartpole environment to introduce stochasticity. That is why I specified "original". Both the stochastic gridworld and the ant environments are also hand-crafted to showcase properties of the algorithm (and not properties of the real world). It makes it difficult to catch the general interest of beta, beyond the considered environments.

We are sorry for our misunderstanding. We modified the environment to showcase two types of stochasticity, one due to the agent's actions, and another due to the environment's reactions. As Reviewer 2 mentions, from the point of view of the agent, the actual source of stochasticity might be hard to distinguish due to their resource limitations, and beta in this case can capture the sensitivity to either an "intrinsic" (ontological) stochasticity (a property of the "real world") or more of an epistemological stochasticity (a property of the knowledge of the agent about the "real world"). Regardless of the source, MOP can model this sensitivity to state-transition uncertainty, as we mention in the 4th paragraph in the Discussion (new paragraph). Please note that we have introduced the notion of risk-sensitivity to talk about why MOP agents change their behavior when there is a stochastic region in the environment.

I still have concerns about the significance of this work. Except for a single citation late in the paper, there are no arguments that support the importance of targeting inherently stochastic areas.

Overall, what does this principle may explain about intelligent behaviors of biological systems that others don't? I don't find a clear answer to that question. Most comparison baselines show keep-alive behaviors, raw variability is not fairly compared and targeting stochastic areas does not sound important.

Thank you for highlight that more arguments for targeting stochastic areas would make our narrative more convincing. We have included a discussion on risk-sensitivity in the new paragraph in the Discussion. As argued by Reviewer 2, going into depth in this would require a series of papers, and indeed we intend to extend MOP to the modeling of risk-sensitivity and entropy-seeking in human behavior.

The authors could as well drop the beta-part from the loss function, remove the evaluation on hand-crafted stochastic environment and leave the paper as it is. It would change nothing with respect to the objective of the paper. But in this case, it results in an already-implemented principle which is not compared to important

previous methods (I'm not convinced by the irrelevance of the comparison). When I say arguments, it could simply be intuitive examples and a discussion of behaviors in biological systems.

As Reviewer 2 suggests, it would be undesirable to remove that beta part of our algorithm, as it pertains to the interesting discussion of whether animals are risk seeking or risk avoiders. We have explained this in our new paragraph in the Discussion.

We would like to add that even with $\beta=0$, the case of $\alpha=1$ with no rewards has not been considered in the literature as a powerful case scenario, with important conceptual implications, as far as we know.

Since the authors addressed none of my two main concerns, raised in the previous iteration (rigorous variability comparison or strong arguments for the importance of targeting stochastic areas), I recommend a reject.

We hope that with the new additions the reviewer finds convincing support for the conceptual leap that our new approach represents to understand and interpret behavior.

Note that in the Discussion we have provided a succinct list of experimental observations to avoid an extremely long discussion

REVIEWERS' COMMENTS

Reviewer #1 (Remarks to the Author):

The authors included the risk-sensitivity suggestion made by R2 and addressed my last concerns. Thus, I'm glad to also recommend the acceptance of the paper.